# MERMAIDE: Learning to Align Learners using Model-Based Meta-Learning

**Arundhati Banerjee**[*]                                                                          *arundhat@cs.cmu.edu*
*School of Computer Science*
*Carnegie Mellon University*

**Soham Phade**                                                                          *soham_phade@berkeley.edu*
*Salesforce*

**Stefano Ermon**                                                                          *ermon@cs.stanford.edu*
*Department of Computer Science*
*Stanford University*

**Stephan Zheng**                                                                          *stephan@asari.ai*
*Asari AI*

**Reviewed on OpenReview:** *https://openreview.net/forum?id=H5VRvCXCzf*

## Abstract

We study how a principal can efficiently and effectively intervene on the rewards of a previously unseen *learning* agent in order to induce desirable outcomes. This is relevant to many real-world settings like auctions or taxation, where the principal may not know the learning behavior nor the rewards of real people. Moreover, the principal should be few-shot adaptable and minimize the number of interventions, because interventions are often costly. We introduce MERMAIDE, a model-based meta-learning framework to train a principal that can quickly adapt to out-of-distribution agents with different learning strategies and reward functions. We validate this approach step-by-step. First, in a Stackelberg setting with a best-response agent, we show that meta-learning enables quick convergence to the theoretically known Stackelberg equilibrium at test time, although noisy observations severely increase the sample complexity. We then show that our model-based meta-learning approach is cost-effective in intervening on bandit agents with unseen explore-exploit strategies. Finally, we outperform baselines that use either meta-learning or agent behavior modeling, in both 0-shot and 1-shot settings with partial agent information.

## 1 Introduction

In many application domains, such as revenue maximization in auctions (Milgrom & Milgrom, 2004), economic policy design for social welfare (Zheng et al., 2022) or optimizing skill acquisition in personalized education (Maghsudi et al., 2021), a principal seeks to incentivize an *adaptive* agent to achieve the principal's goal. In this work, we assume that both the principal and the agent are learners and the principal incentivizes by directly intervening on the rewards of the agent. However, the principal does not know neither the exact value of the agent's rewards nor its learning algorithm (or its parameters). For instance, a government (principal) may want to incentivize the use of environmentally-friendly products by levying green taxes, but needs to consider that people (agents) may change their consumption behavior as taxes change. Here, common agent models based on rationality or forms of bounded rationality often do not fully describe real-world behavior. Hence, interacting with the agents is required for the principal to *learn* how agents change their behavior, but such interactions are not "free" or without risk. For example, measuring the impact of a tax change

---
[*]Work done during an internship at Salesforce

on consumers takes effort, while it may incur economic costs if the new tax is unfair or not well calibrated. In a similar vein, for designing intelligent tutoring systems, the tutor (principal) has to adapt to different students (agents) and learn to incentivize the students to understand different concepts, without the designed curriculum being too difficult or too easy (incurring cost for the principal). In personalized health-monitoring apps, the app (principal) has to adapt to the user's (agent) lifestyle and preferences to maybe create a plan that can incentivize the user to fulfill their health-related goals.

To reduce the need for real-world interactions, we can use simulations with deep reinforcement learning (RL) agents. This is an attractive solution framework: deep neural networks are expressive enough to imitate real-world entities and simulations can be run safely and at will. Moreover, we can use deep RL to learn intervention policies that are effective even in the face of complex agent behaviors in *sequential* principal-agent problems.

However, this approach also faces several challenges. When deploying the learned policies in the real world, interventions can typically only be applied a few times, due to implementation costs, and rarely under identical circumstances; in contrast to simulations, we cannot reset the real world. Even though principals may adapt their policies to new conditions, they cannot realistically know the true rewards or learning strategy of the agent. Hence, our goal is to learn the principal's intervention policy that 1) can perform well even when agents learn, 2) can be quickly adapted, 3) is robust to distribution shifts in agent behaviors, and 4) is effective despite having only *partial information*.

**Contributions.** To address these challenges, we propose MERMAIDE (**Me**ta-lea**r**ning for **M**odel-based **A**daptive **I**ncentive **De**sign), a deep RL approach that 1) learns a world model and 2) uses gradient-based meta-learning to learn a principal policy that can be quickly adapted to perform well on unseen test agents. We consider a *principal and single-agent* setting wherein the principal intervenes *at a cost* on the agent's learning process to incentivize the agent to learn to act to achieve the principal's objective. We assume that the agent behaves in a first-order strategic manner and the principal in a second-order strategic manner. Here, the agents optimize their experienced rewards and minimize their regret, but do not account for their influence on the principal's actions. In contrast, the principal intervenes explicitly as to influence the agent's actions.

**Claims.** In summary, our work demonstrates the challenges faced in the principal and single agent setup with adaptive learning agents. MERMAIDE is an effective framework to address these challenges.

Our empirical results validate the advantages, but also show the limitations of a purely model-based approach to address the non-stationarity in the environment for the principal and the adaptive agents. Our proposed solution, MERMAIDE, shows that model-based meta-learning is able to further improve on a purely model-based approach and learns a cost-effective few-shot adaptable intervention policy for the principal.

More specifically, in this work, we empirically verify the following claims about MERMAIDE:

**Claim 1.** In a single-round Stackelberg game between the principal and an agent, where the principal (leader) acts first by deciding whether to intervene or not, and the agent (follower) acts second according to a best-response policy (Von Stackelberg, 2010), our meta-trained principal reliably finds solutions that one-shot adapt well with a best response agent under both perfect and noisy observations for the agent and the principal. Furthermore, the principal's out-of-distribution performance depends on its observable information about the agent.

**Claim 2.** In the multi-armed bandit setting, MERMAIDE finds well-performing reward intervention policies in a repeated interactive setup with the adaptive bandit agents. MERMAIDE's test-time performance and robustness against out-of-distribution bandit learners depends on the agent's level of exploration and their level of pessimism in the face of uncertainty, which are unknown to the principal. In particular, this holds for both $K = 0$-shot and $K = 1$-shot evaluation.

To the best of our knowledge, prior works studying the principal-agent setup, even in an adaptive setting, overlook several of these problems with more restrictive assumptions on either the observations available or the types of agents and principal.

## 2 Related Work

**Mechanism design.**  The principal-agent problem (Eisenhardt, 1989) studies how a principal can incentivize and align the agent with the principal's goals, in particular in the face of information asymmetry (e.g., the principal does not know the agent's rewards). This relates to the general mechanism design setting in which a principal typically interacts with multiple agents (Hurwicz & Reiter, 2006). Learning a mechanism with agents who also learn is a bilevel optimization problem, which is NP-hard (Ben-Ayed & Blair, 1990; Sinha et al., 2017). Possible solution techniques include branch-and-bound and trust regions (Colson et al., 2007). In particular, solving bilevel optimization using joint learning of the mechanism and the agents can be unstable, since agents continuously adapt their behavior to changes in the mechanism. This can be stabilized using curriculum learning (Zheng et al., 2020), but generally bilevel problems remain challenging, especially with nonlinear objectives or constraints. Moreover, unlike with typical curriculum learning (Bengio et al., 2009), in our setting, the principal's intervention essentially changes the task for the adaptive agent, thus presenting a non-stationary learning problem for both the principal and the agent at each time step.

**Adaptive mechanism design.**  Previous work in mechanism design usually does not consider learning how to learn to incentivize across agents of different types. Pardoe et al. (2006) found that a form of meta-learning that adapts the learning process itself can design English auctions (sequential bidding) that perform better with adaptive bidders who are loss-averse, and is still effective when the distribution of bidder behaviors (slowly) shifts. Prior work has studied algorithms for incentivizing exploration in bandit agents (Chen et al., 2018; Wang et al., 2021) where the principal can assign incentives to temporary and myopic agents for choosing different arms so that the principal can passively determine the global preferences of the agent population. Shi et al. (2021) extend this line of work to consider non-myopic strategic agents, but unlike our principal-agent problem formulation, they assume that the agent is aware of the principal's incentive before choosing an arm and the agent always selects the incentivized arm. Moreover, their setting does not focus on few-shot generalization to unseen test agents. Our work expands on this theme by explicitly modeling agents that learn, considering shifts in the *learning algorithm* of the agents, and using deep RL with meta-learning. This combination enables learning incentives that generalize well across more complex tasks.

**Meta-learning and inverse RL.**  In recent years, gradient-based meta-learning has proven effective in learning initializations for complex policy models that generalize well to unseen tasks (Finn et al., 2017a; Nagabandi et al., 2018). Luketina et al. (2022) study meta-gradients for adapting in environments with controlled sources of non-stationarity, but ignore non-stationarity from interactions between strategic agents that learn. Prior works in imitation learning (Argall et al., 2009) and inverse RL (Abbeel & Ng, 2004) assume access to expert demonstrations with a fixed policy to imitate or learn the reward function of, whereas Jacq et al. (2019); Ramponi et al. (2020) consider inverse RL with observers from learners that improve their policies, but they do not feature a principal that actively intervenes. In contrast, our principal aims to learn a policy that can strategically *alter* the behavior of such demonstrators (our agents), who are themselves *learning* during an episode of the demonstration. Recently, Boutilier et al. (2020) studied meta-learning for bandit policies, while Guo et al. (2021) introduced the inverse bandit setup for learning from low-regret demonstrators. However, these works do not consider shifts in the bandit learning algorithm between training and test time.

**Modeling agents.**  A key challenge in multi-agent learning is that each agent experiences a non-stationary environment if other agents are learning. As such, agents can benefit from having a *world model*, e.g., to know what the policy or value function of the other agents are. World models can stabilize multi-agent RL (Lowe et al., 2017) and enable higher-order learning methods (Foerster et al., 2018), and are a form of model-based RL. However, this may require a large amount of observational data or prior knowledge, which may be hard to acquire. We show that world models make principals much more efficient in our setting.

## 3 Problem formulation

**Overview.** We model a *principal* who aims to incentivize an *agent* to (learn to) execute the principal's preferred action. To do so, the principal can *intervene* and change the agent's rewards at a cost. Without interventions, the agent may learn to prefer an action different than the principal's.

For example, consider consumers who can use either environmentally "clean" or "dirty" goods. Indifferent at first, consumers may gradually learn to prefer dirty goods if those are consistently cheaper than clean ones, whereas the government may want them to prefer clean goods. Here, the agent's reward is the negative of the cost of consumption, for instance, and an intervention changes the price of goods through taxes or subsidies. If we can use a simulation, the principal can compute an optimal intervention. However, the simulation might be inaccurate and real-world agents might behave differently. As an example of such test-time distribution shift, simulated agents may be quick to change their consumption patterns, while real agents may be slow. A "good" principal (trained in a simulation) could quickly be fine-tuned to intervene more in the latter case and adapt quickly if such behavior is observed during deployment.

In particular, we focus on *learning* a principal policy that can adapt quickly at test-time (e.g., deploying taxes and subsidies in the real world), and that is effective when the agent's learning algorithm differs from that during training.

We now formalize this setting. In this work, we focus on agents in a stateless environment for ease of exposition. For all variables and their meaning, see Appendix A.

**The agent.** The agents are characterized by their action space $A$ and a base reward function $r : A \to \mathbb{R}$. We call it base reward as the agent experiences an *intervened* reward

$$\tilde{r}_t(a_t) = r(a_t) + r'_t(a_t), \tag{1}$$

where the intervention $r'_t$ is provided externally (by the principal) for the agent action $a_t$. We index time as $t = 1, \ldots, T$. At each time step $t$, the agent's policy $\pi_t$ computes a distribution over its actions based on the observations for the agent up to timestep $t$ and executes $a_t \sim \pi_t$. We assume that the principal has a *preferred action* $a^* \in A$ that the agent should execute, whereas the agent's optimal policy can prefer a different action than $a^*$ without intervention. Finally, at time $t$, the agent learns using an update rule $f : (\pi_t, a_t, \tilde{r}_t) \mapsto \pi_{t+1}$ to maximize the agent's intervened rewards, e.g., under UCB (Lai et al., 1985), $f$ updates the confidence bounds for the action selected at time $t$.

**The principal.** In this work, from the principal's point of view, the *world (environment)* consists of the agent who maximizes $\tilde{r}$. A standard assumption is that agents are rational and they may have a private state (referred to as its *type*) which the principal cannot see. Although the agent faces a stateless problem, *the principal faces a stateful problem with partial observability*. The full state $s \in S$ includes the principal's internal state $h_t^p$ (e.g., the principal's belief about the value of the private agent information), and all information about the agent, including its past actions, reward function, and policy model; often, the latter two are private.

More formally, the principal can be modeled as a POMDP $(S, o^p, A^p, r^p, \gamma, \mathcal{P})$. It receives observations $o^p$ (a part of the world state $s$), $A^p$ is its action space of interventions, $r^p$ is its reward, $\gamma$ is a discounting factor, and $\mathcal{P}$ are the environment dynamics, e.g., as caused by the agent's actions. At time $t$, the principal samples an action $\boldsymbol{a}_t^p \sim \pi^p\left(\boldsymbol{a}_t^p | o_{t-1}^p, h_{t-1}^p\right)$ which determines its intervention on each possible agent action $a$, i.e. $\boldsymbol{a}_t^p = \left[r'_1, \ldots, r'_{|A|}\right]$.

**Adaptive intervention policy learning** To model distribution shift at test time, we follow the meta-learning terminology (Finn et al., 2017b) and view each distinct agent as a *task* $\tau^i$. The principal has access to a *train set of agents* $\tau^i \in \mathcal{T}_{\text{train}}; i = 1, \ldots, n_{\text{train}}$ and is evaluated on a *test set of agents* $\tau^i \in \mathcal{T}_{\text{test}}; i = 1, \ldots, n_{\text{test}}$. *We emphasize that during a task, both the principal and agent may learn and adapt, both at train and test time.*

Here, we focus on two key challenges: $K$-shot adaptation and distribution shift. First, the principal gets only $K$ episodes for fine-tuning for each test task (but can train indefinitely for each train task). Second, the principal faces two types of distribution shift: 1) across tasks and 2) intra-task non-stationarity. The train and test tasks may differ (significantly) in their temporal distribution of actions, e.g., due to different agent updates $f$ or the agent rewards $r_t$ being centered around different values (e.g., average price levels are higher in the real world vs in the simulation). Within a task, the agent's learning is affected by the principal's interventions that change its reward $\tilde{r}$. This gives rise to non-stationarity in the agent's environment, as its learning objective may shift over time. These forms of distribution shift distinguish our adaptive intervention policy learning setting from most prior work in meta-learning, which often assume stationarity within a task and also assume similar task distributions at train and test times.

**Objectives.** The principal's objective is to maximize how often test-time agents choose $a^*$ during learning and have them converge to a policy that always chooses $a^*$. To do so, the principal aims to maximize the cost-adjusted test-time return $J_{\text{test}}^p \left( \pi^p, \pi^i \right) = \sum_{t=1}^{T} \gamma^{t-1} (r_t^p - \alpha c_t)$, where the agent executes its (optimal) policy $\pi^i [\pi^p]$ in response to $\pi^p$ and the principal incurs a cost $c_t$ if it intervenes. $r_t^p = \mathbf{1}\left[ a_t = a^* \right]$, $\alpha > 0$.

$$\pi^{p*} = \arg\max_{\pi^p} \mathbb{E}_{\tau^i \in \mathcal{T}_{\text{test}}} \mathbb{E}_{\pi^p} \mathbb{E}_{\pi^i [\pi^p]} \left[ \sum_{t=1}^{T} \gamma^{t-1} (r_t^p - \alpha c_t) \right] \tag{2}$$

A simple cost function is $c_t = \mathbf{1}\left[ r' \neq 0 \right]$, i.e., the cost is constant across non-trivial interventions, where $\alpha > 0$ is a constant. Note that if intervention were free ($c_t = 0$), a trivial solution is to always add a large $r'(a^*) \gg 0$ for its preferred action $a^*$, such that it always yields the highest reward. Hence, we focus on learning non-trivial strategies when intervention is costly, which forces the principal to strategically alter the agent's learning behavior.

During an episode of $T$ time steps, each agent $i$ starts with a uniformly initialized action probability distribution $\pi_0^i$ and optimizes $\pi_t^i$ subject to interventions $\pi^p$ to maximize its return: $\mathbb{E}_{\pi^i} \mathbb{E}_{\pi^p} \left[ \sum_{t=1}^{T} \tilde{r}_t^i \left( a_t^i, a_t^p \right) \right]$. Here, we assume that $T$ and $\gamma$ are sufficiently large so the agent converges to its optimal policy under $\tilde{r}$, using its learning algorithm $f$. That is, we assume that the objective in Equation (2) is sufficient to describe the principal's objective of ensuring the agent converges to preferring $a^*$ at some $t < T$.

In the $K$-shot adaptation setting, at test time, the principal gets $K$ episodes to interact with any agent, each episode of length $T$ steps. The principal has a fixed policy during an episode and it can update its policy at the end of an episode. The agent is reset across episodes, and within each episode, the agent follows its own learning strategy in response to the principal's interventions. On the $K + 1^{\text{th}}$ episode, the principal evaluates its $K$-shot adapted policy on the agent. Note this assumes that the principal has a separate copy of the test time agent for evaluation.

## 4 MERMAIDE: Learning to Align Learners

MERMAIDE learns an intervention policy to align the agent's preferred action with the principal's one, using:

1) a recurrent *world model* parameterized by $\omega$ that outputs a distribution over an agent $i$'s actions at the next time step $t$: $\hat{\pi}_\omega \left( a_t^i | a_{t-1}^i, a_{t-1}^p, h_{t-1}^i \right)$, conditioned on the planner's intervention and the observed agent action at $t-1$. $h_{t-1}^i$ is the hidden world model state. $a_t^i \sim \pi_t^i$.

2) a recurrent *intervention policy* which outputs a distribution over interventions $a_t^p \sim \pi_\theta^p \left( a_t^p | a_{t-1}^i, a_{t-1}^p, \hat{a}_t^i, h_{t-1}^p \right)$, conditioned on its previous intervention, the observed agent action and the world model's predicted next agent action $\hat{a}_t^i = \max_a \hat{\pi}_\omega \left( a | a_{t-1}^i, a_{t-1}^p, h_{t-1}^i \right)$. $h_{t-1}^p$ is the hidden state of the policy network.

We train this with gradient-based meta-learning and RL (Algorithm 1). $\mu^i$ indicates the mean or base reward function for agent $i$ and $\tilde{\mu}^i$ is the reward function after the principal's intervention $a_t^p$ at time $t$. Please refer to Table 3 and Table 4 for a comprehensive list of the notations used. Here, the principal maximizes

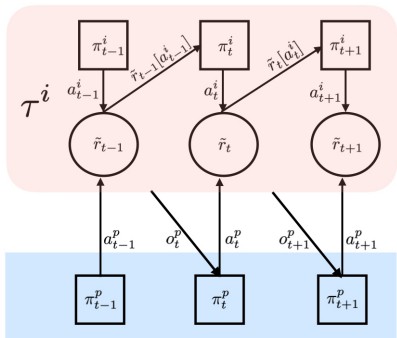 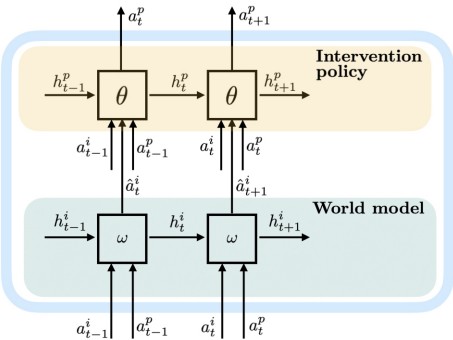

Figure 1: Overview of **MERMAIDE.** Left: Flow of principal and agent observables, rewards, and actions. Right: The principal's world model and intervention policy. Also see Algorithm 1.

---

**Algorithm 1** MERMAIDE (Notations also in Table 4)

1: Initialize principal $(\theta_0, \omega_0)$, and hidden states $h_0^i, h_0^p$.
2: **for** meta-train epoch $e = 1, \ldots, E_{\text{train}}$ **do**
3:     Update world model parameters $\omega = \omega_e$ (Equation (3)).
4:     **for** agents (tasks) $i = 1, \ldots, n_{\text{train}}$ **do**
5:         Initialize agent: $(\mu^i, \pi_0^i)$, task specific principal policy parameter $\theta\left(\tau_0^i\right) = \theta_e$.
6:         **for** $k = 1, \ldots, K_{\text{train}}$ **do**
7:             **for** time t $= 1, \ldots, T$ **do**
8:                 Predict $\hat{a}_t^i = \arg\max_{a_t^i} \hat{\pi}_\omega \left(a_t^i | a_{t-1}^i, a_{t-1}^p, h_{t-1}^i\right)$
9:                 Intervention: $\tilde{\mu}^i = \mu^i + a_t^p, \quad a_t^p \sim \pi_{\theta\left(\tau_k^i\right)}^p \left(a_t^p | a_{t-1}^i, a_{t-1}^p, \hat{a}_t^i, h_{t-1}^p\right).$
10:                 Agent acts: $a_t^i \sim \pi_t^i$ and receives reward $r_t^i \sim \mathcal{N}\left(\tilde{\mu}^i, \sigma^2\right)$. $\pi_t^i \mapsto \pi_{t+1}^i$.
11:             **end for**
12:             Locally update $\theta\left(\tau_k^i\right) \mapsto \theta\left(\tau_{k+1}^i\right)$. {Using REINFORCE.}
13:         **end for**{Rollout for meta-update; $\mathcal{D}_{\text{meta}}\left(\tau^i\right) = \{\}$}
14:         **for** $t = 1, \ldots, T$ **do**
15:             Predict $\hat{a}_t^i = \arg\max_{a_t^i} \hat{\pi}_\omega \left(a_t^i | a_{t-1}^i, a_{t-1}^p, h_{t-1}^i\right)$
16:             Intervention: $\tilde{\mu}^i = \mu^i + a_t^p, \quad a_t^p \sim \pi_{\theta\left(\tau_{K_{\text{train}}}^i\right)}^p \left(a_t^p | a_{t-1}^i, a_{t-1}^p, \hat{a}_t^i, h_{t-1}^p\right).$
17:             Agent acts: $a_t^i \sim \pi_t^i$, receives reward $r_t^i \sim \mathcal{N}\left(\tilde{\mu}^i, \sigma^2\right)$. Updates $\pi_t^i \mapsto \pi_{t+1}^i$.
18:             Collect $\mathcal{D}_{\text{meta}}\left(\tau^i\right) \cup \left\{a_t^i, a_t^p, \pi_{\theta\left(\tau_{K_{\text{train}}}^i\right)}^p\right\}$
19:         **end for**
20:     **end for**
21:     Meta-update $\theta_e \mapsto \theta_{e+1}$ using $\mathcal{D}_{\text{meta}} = \cup_{\tau^i} \mathcal{D}_{\text{meta}}\left(\tau^i\right)$. {Using MAML.}
22: **end for**

---

$J_{\text{train}}^p$ similar to the objective in Equation (2). The base RL algorithm is REINFORCE (Williams, 1992) and the meta-learning update uses MAML (Finn et al., 2017b). The agent optimizes its cumulative intervened reward (see Section 6). The world model $\hat{\pi}_\omega$ trains by maximizing the log-likelihood of the observed $a_t^i$, using Adam (Kingma & Ba, 2014):

$$\arg\max_\omega \mathbb{E}_{a^p} \mathbb{E}_{a^i} \left[\sum_{t=1}^T \log \hat{\pi}_\omega \left(a_t^i | a_{t-1}^i, a_{t-1}^p, h_{t-1}^i\right)\right] \tag{3}$$

Note that the principal's parameters $\theta$ are updated after each $T$-step episode, while the agent continuously learns during each episode. Also, the agent is reset in between episodes. At time 0, the world model makes a prediction based on zero initialization. We use a single world model for all agents. At meta-test time, only the intervention policy is updated by one-shot adaptation to a new agent (Algorithm 2).

# 5 Evaluating Meta-Learning for the Principal

We now compare the learning behavior of a principal that is meta-trained (MAML) (Finn et al., 2017b) versus one trained with standard policy gradients (RL), in a simple Stackelberg game between a principal and an agent. The Stackelberg game considers a leader-follower approach where the principal (leader) acts first, deciding to intervene or not on the agent (follower) who acts second according to a best-response policy. The key characteristic of a Stackelberg equilibrium is that once the leader has made their move, the followers have no incentive to deviate from their optimal responses, given the leader's action. This results in a stable, strategic outcome where each player is maximizing their utility or payoff based on the actions of their opponent. Our goal is to learn an intervention policy for the principal that can adapt to different agent types and find the Stackelberg equilibrium.

The agent's actions are "cooperate" and "defect", while the principal can choose whether or not to intervene. Assuming the row player is the agent and the column player is the principal, we define a $2 \times 2$ payoff matrix

$$
\begin{array}{cc}
 & \text{No intervention (NI)} \quad \text{Intervene (IN)} \\
\begin{array}{c} \text{Cooperate (C)} \\ \text{Defect (D)} \end{array} &
\begin{pmatrix}
u, 1 & u+1, 1-c \\
1-u, 0 & -u, -c
\end{pmatrix},
\end{array}
$$

where $u \in (0, 1)$ and $c$ is the cost of intervention ($c < 1$). The agent's base payoff $u$ is its type. The principal prefers cooperation: it gets 1 if the agent cooperates and 0 if the agent defects (minus the cost $c$ if it intervenes). Note that intervention incentivizes cooperation ($u + 1 > -u$).

We now analyze three scenarios with increasing difficulty:

1) First, we assume that the principal knows $u$. Here, there is a unique Stackelberg equilibrium at (C, NI) when $u \geq \frac{1}{2}$, and at (C, IN) when $u < \frac{1}{2}$.

2) Second, the principal observes a noisy version of $u$. In both cases, the agent first observes the principal's action and plays its best-response (knowing the payoffs).

Note that both 1) and 2) are single-round Stackelberg settings.

3) Finally, we study a repeated (multi-round) Stackelberg setting where the agent cannot see the principal's action. Instead, we assume that the agent keeps a running average for the experienced payoffs for each of its actions. In an equivalent single-round setting this would correspond to the principal committing to a mixed action and then the agent choosing its best response. When $u \geq \frac{1}{2}$, the Stackelberg equilibrium occurs at (C, NI). When $u < \frac{1}{2}$, at the Stackelberg equilibrium the principal has a mixed action where it chooses to intervene for $\frac{2u+1}{2}$ fraction of times and the agent always cooperates.

Given these expected Stackelberg equilibrium strategies, our goal is to learn a policy for the principal that predicts its probability of intervention and study the adaptivity of MAML vs RL trained policies on unseen agents. We set $c = 0.75$. Given a set of training agents with different types $u \sim \mathcal{U}(0, 1)$, the principal learns the optimal policy parameters $\theta^* = \arg\max_\theta \mathbb{E}_u \mathbb{E}_{a^p \sim \pi_\theta^p(u)} [r^p(a^p)]$. Note that here we do not use a world model, rather we focus on model-free policy learning. We study the quality of the initialization $\theta_{\text{MAML}}$ vs $\theta_{\text{RL}}$ by evaluating the one-shot adaptation performance of the trained principal on unseen test agents.

**With perfect observability.** In this setting, we assume that the principal observes an agent's exact payoff parameter $u$. Figure 2 shows the principal's meta-test time probability of intervening with 3 different agents from the test set, across training epochs. The principal and agent should be at different Stackelberg equilibria depending on the type $u$, as discussed above. We see that a principal trained from scratch on the test agents using standard policy gradients is unable to adapt to different agents in a single-shot adaptation setting. In contrast, with meta-learning, the principal learns a policy $\pi_{\theta_{\text{MAML}}}^p$ that is one-shot adaptable to agents of different types and converges to the correct Stackelberg equilibrium at meta-test time.

**With noisy observations for the principal.** Here, we emulate a principal with partial observability of the agent, by letting the principal observe $u$ with added i.i.d. Gaussian noise. The agent can see all

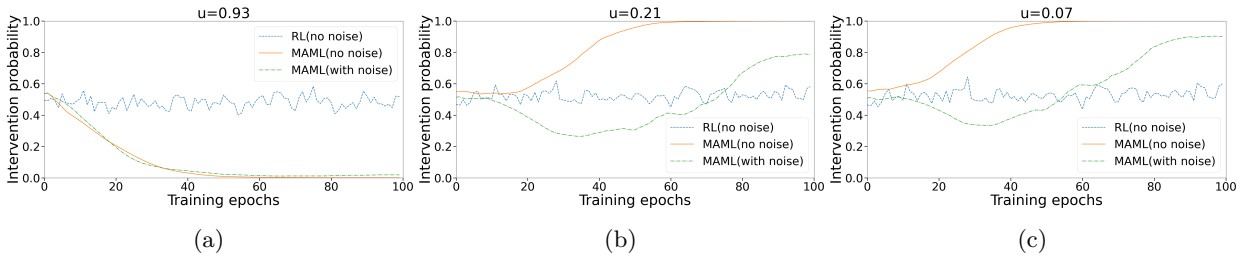

Figure 2: **Single round game.** REINFORCE (RL) does not adapt to expected Stackelberg equilibrium during evaluation. MAML's adaptability suffers under observation noise.

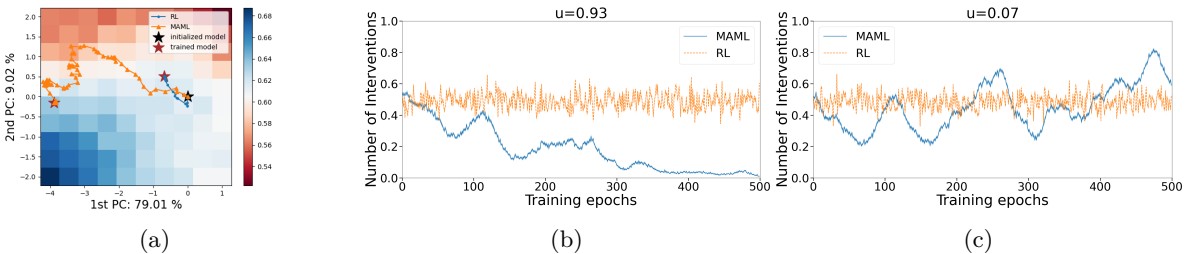

Figure 3: **Multi-round game.** (a) Principal's optimization trajectory in the expected payoff landscape during training. Axes are PCA directions in the policy parameter space. Colorbar indicates the principal's expected payoff over the training agents. (b)(c) MAML adapts (single shot) to Stackelberg equilibrium with a best response agent. The number of interventions are normalized by the episode length $T = 100$.

payoffs and chooses the best response to achieve a Stackelberg equilibrium. Figure 2 shows that with noisy observations, the meta-learned principal policy requires more training time to be one-shot adaptable to the equilibrium intervention policy. This empirically indicates the increased difficulty of learning an adaptive intervention policy due to incomplete information about the agent, especially under limited adaptation time with unseen agents. It therefore motivates us to adopt a *model-based* approach for the principal to better estimate the agent type and learn an adaptive intervention policy.

Comparing Figure 2b and Figure 2c, when $u < \frac{1}{2}$, the difference in unintervened payoffs between the principal's preferred action ($u$) and the agent's preferred action ($1 - u$) also impacts the one-shot adaptability of the principal with noisy observations. This informs our analysis in Section 6.

**Multi-round repeated game with noisy rewards.** In this setting, the principal and agent repeatedly play an iterated game over $T = 100$ steps. In each round, the principal observes the agent's type $u$ with added i.i.d. Gaussian noise. The agent cannot see the principal's actions, and best responds given its payoff estimates. Whenever the agent selects an action, it receives a noisy observation of the true payoff and updates its estimate. Compared to the single-round setting, here the agent's best response behavior may change across rounds in the game depending on its observed payoffs, giving rise to non-stationarity in the principal's environment. The planner, in turn, has to learn to intervene so that the agent's best response is to cooperate.

Figure 3a compares the optimization trajectory followed using 1) policy gradients and 2) meta-learning for the principal. Starting from the same initialization, the meta-learned policy's parameters lie in a region of the payoff landscape with a higher expected value over the training agents. Moreover, Figure 3b and Figure 3c show the one-shot adaptability of the principal's policy for two different agent types at meta-test time. Meta-learning helps learn a better intervention strategy that is robust to the principal's observation noise as well as the agent's evolving best response strategy.

This analysis shows the benefits of meta-learning over standard policy gradients in learning a principal's intervention policy at equilibrium with strategic agents and under partial observability for the principal in a simple Stackelberg setting. Our observations motivate using meta-training and model learning in general

Table 1: **Test-time principal performance on agents with different test-time learning parameters (3 random seeds).** Left column: Principal's algorithm (e.g., MERMAIDE), training agent type (e.g., UCB with $\beta = 0.17$). Other columns: Test-time scores (mean and standard error (s.e.)) on agents with the same algorithm, but different hyperparameters. MERMAIDE ($K = 0$) indicates zero-shot evaluation. Rest are evaluated with one-shot adaptation ($K = 1$). MERMAIDE outperforms all baselines in almost all cases in both the $K = 0$ and $K = 1$ cases, with $K = 1$ principals doing better than $K = 0$.

| **Train on UCB,** $\beta = 0.17$ | Test on $\beta = 0.17$ | $\beta = 0.27$ | $\beta = 0.42$ | $\beta = 0.5$ | $\beta = 0.67$ |
|---|---|---|---|---|---|
| *No intervention* | 3 (0) | 5 (0) | 8 (0) | 10 (0) | 12 (0) |
| MF-RL | 119 (2) | 109 (2) | 98 (2) | 90 (2) | 77 (1) |
| MF-MAML | 133 (2) | 125 (3) | 107 (1) | 97 (1) | 77 (0) |
| WM-RL | 123 (7) | 112 (6) | 100 (4) | 92 (2) | 75 (1) |
| MERMAIDE ($K = 1$) | **154 (2)** | **151 (1)** | **129 (1)** | **129 (1)** | **87 (0)** |
| MERMAIDE ($K = 0$) | 148 (2) | 138(1) | 120(1) | 103(2) | 89(1) |
| **Train on $\epsilon$-greedy,** $\epsilon = 0.1$ | $\epsilon = 0.1$ | $\epsilon = 0.2$ | $\epsilon = 0.3$ | $\epsilon = 0.4$ | $\epsilon = 0.5$ |
| *No intervention* | 3 (0) | 4 (1) | 7 (0) | 9 (1) | 11 (0) |
| MF-RL | 115 (5) | 94 (4) | 54 (19) | 39 (6) | 22 (9) |
| MF-MAML | 122 (4) | 97 (3) | 58 (5) | 40 (2) | 12 (1) |
| WM-RL | 115 (4) | 94 (5) | 70 (1) | 55 (3) | **38 (1)** |
| MERMAIDE ($K = 1$) | **138 (1)** | **112 (2)** | **85 (3)** | **66 (2)** | 37 (4) |
| MERMAIDE ($K = 0$) | 133 (2) | 109 (3) | 86 (2) | 65 (3) | 37 (1) |
| **Train on UCB,** $\beta = 0.67$ | $\beta = 0.17$ | $\beta = 0.27$ | $\beta = 0.42$ | $\beta = 0.5$ | $\beta = 0.67$ |
| *No intervention* | 3 (0) | 5 (0) | 8 (0) | 10 (0) | 12 (0) |
| MF-RL | 103 (3) | 101 (3) | 92 (2) | 85 (1) | 74 (1) |
| MF-MAML | 124 (2) | 116 (1) | 102 (1) | 94 (1) | 80 (1) |
| WM-RL | 100 (4) | 89 (0) | 85 (1) | 85 (1) | 74 (0) |
| MERMAIDE ($K = 1$) | **132 (1)** | **130 (1)** | **123 (2)** | **115 (2)** | **99 (1)** |
| MERMAIDE ($K = 0$) | 115 (3) | 114 (3) | 103 (4) | 100 (3) | 89 (3) |
| **Train on $\epsilon$-greedy,** $\epsilon = 0.5$ | $\epsilon = 0.1$ | $\epsilon = 0.2$ | $\epsilon = 0.3$ | $\epsilon = 0.4$ | $\epsilon = 0.5$ |
| *No intervention* | 3 (0) | 4 (1) | 7 (0) | 9 (1) | 11 (0) |
| MF-RL | 4 (5) | 2 (3) | 5 (0) | 11 (5) | 7 (1) |
| MF-MAML | 2 (0) | 4 (0) | 6 (0) | 8 (1) | 11 (1) |
| WM-RL | 102 (6) | 79 (10) | 68 (3) | 47 (1) | 30 (2) |
| MERMAIDE ($K = 1$) | 87 (42) | **102 (3)** | **78 (6)** | **69 (1)** | **46 (2)** |
| MERMAIDE ($K = 0$) | 113 (20) | 85 (15) | 71 (16) | 48 (14) | 21 (15) |

principal-agent settings. MERMAIDE uses both these components; next, we analyze its performance against learning agents with complex adaptive behavior under repeated interventions.

## 6 Evaluating MERMAIDE on Bandits

We now consider a principal intervening sequentially on an adaptive no-regret learner agent, modeled by an $|A|$-armed bandit instance with action set $A$ having base reward $\boldsymbol{r} = \left[r_1, \ldots, r_{|A|}\right]$. At each time step $t$, the agent chooses an arm $a$ and gets a reward sampled from $\mathcal{N}\left(r_a, \sigma^2\right)$. We assume $r_a \in (0, 1)\ \forall a$. The agent aims to maximize its cumulative reward over a horizon of $T$ steps. The agent can only observe the reward for the chosen action, and hence faces a explore-exploit dilemma addressed by bandit algorithms like UCB (Lai et al., 1985). We assume there is a unique arm $\tilde{a}$ with the highest base reward: $\tilde{a} = \arg\max_a r_a$, i.e., the agent's preferred action without any intervention.

**Costly interventions.** To analyze the effect of the cost of intervention $c_t$ on the principal's learnt policy, we assume that the principal decides among three different intervention levels $|r'| \in \{0, 0.5, 1\}$ such that $c_t = |r'|$. Across different bandit agent tasks $\tau^i$ with distinct base rewards $\boldsymbol{r}^i$ and reward gaps $\delta = \max_{a \in A} \boldsymbol{r}^i[a] - \boldsymbol{r}^i[a^*]$,

Table 2: **$K$=0-shot evaluation across different agent algorithms: test-time principal scores (3 random seeds).** Left column: principal's algorithm (e.g., MERMAIDE), training agent type (e.g., UCB, $\beta = 0.42$). Other columns: Test-time scores (mean and s.e.) on agents with different algorithm and hyperparameters.

| **Train on UCB,** $\beta = 0.42$ | Test on $\epsilon = 0.1$ | $\epsilon = 0.2$ | $\epsilon = 0.3$ | $\epsilon = 0.4$ | $\epsilon = 0.5$ |
|---|---|---|---|---|---|
| *No intervention* | 3 (0) | 4 (1) | 7 (0) | 9 (1) | 11 (0) |
| WM-RL | 91 (4) | 62 (8) | **68 (1)** | **28 (4)** | - |
| MERMAIDE (ours) | **103 (1)** | **67 (2)** | 30 (2) | 8 (1) | - |
| **Train on $\epsilon$-greedy,** $\epsilon = 0.3$ | Test on $\beta = 0.17$ | $\beta = 0.27$ | $\beta = 0.42$ | $\beta = 0.5$ | $\beta = 0.67$ |
| *No intervention* | 3 (0) | 5 (0) | 8 (0) | 10 (0) | 12 (0) |
| WM-RL | 127 (7) | 95 (2) | 80 (5) | 80 (5) | 61 (4) |
| MERMAIDE (ours) | **138 (2)** | **102 (6)** | **116 (2)** | **96 (5)** | **77 (2)** |

the principal should learn to appropriately incentivize the agent while minimizing the total cost of intervening. We then define the experienced reward as: $\forall a \neq a^*, (a, a^* \in A)$

$$\tilde{r}_t[a^*] = r^i[a^*] + r'_t; \quad \tilde{r}_t[a] = r^i[a] - r'_t. \tag{4}$$

Note that this ensures the agent always experiences an intervention, no matter which action it chooses. During each episode, the agent learns but the principal's policy is fixed; the principal can update its policy only at the end of each episode (Algorithm 1). Also, we assume that the principal can only observe the agent's actions $a_t^i$ but not its base reward $r^i$ or policy update function $f^i$. We measure the performance of the principal using Equation (2), with $\gamma = 1$.

**World model.** The world model predicts the agent's next *action* (given the principal's prior observations) to characterize the agent's behavior. We do not train the principal's world model to estimate the base rewards, because bandit agents with distinct base rewards could still execute the same sequence of actions, depending on the agent's explore-exploit algorithm and its observations.

**Challenges in the sequential setting.** Compared to the setting in Section 5, learning to intervene on sequential (bandit) learners (Appendix B.1) creates more challenges:

1) Bandit agents may use different strategies to maximize their experienced reward. The agent's rate of exploration may be constant (e.g., $\epsilon$-greedy) or it can reduce with time (e.g., UCB) *within an episode*. This creates a highly non-stationary environment for the principal: its intervention policy must adapt to different intra-episode explore-exploit behaviors for the same agent. When the agent explores a larger action space, it further exacerbates these challenges as the principal only has partial information about the agent.

2) Bandit agents are sequential learners and feedback $(a_t^i, \tilde{r}_t^i)$ can update the policy $\pi^i$ differently at different $t$. This may depend on how optimistic (e.g., UCB) or pessimistic (e.g., EXP3) the agents are about their reward estimates. Hence, an intervention $a^p$ may change agent behavior differently at different $t$. Since interventions have different costs, a strategic principal must decide *when* to intervene and *how much* ($|r'|$) based on its observations of the agent's actions.

Appendix B.3 further illustrates the difficulty of this setting.

**Results.** Here, we use 15 bandit agents for training and 10 bandit agents for testing, each with different base rewards (both within and across train and test sets). $|A| = 10$. We consider two agent learning algorithms (UCB and $\epsilon$-greedy) and a range of exploration vs exploitation characteristics, determined by their exploration coefficients: $\beta \in \{0.17, 0.27, 0.42, 0.5, 0.67\}$ for UCB (higher $\beta$ gives more exploration) and $\epsilon \in \{0.1, 0.2, 0.3, 0.4, 0.5\}$ for $\epsilon$-greedy (higher $\epsilon$ gives more exploration). These constants were chosen such that they yield, on average, the same number of exploratory actions with either UCB or $\epsilon$-greedy without any intervention (see Appendix B.2).

Table 1 shows the one-shot adapted principal's score on each test set over $T = 200$ time steps. We compare MERMAIDE with 1) model-free baselines (MF-RL using REINFORCE, MF-MAML using MAML) and 2) REINFORCE with world model (WM-RL); see Appendix B.4. We also include zero-shot evaluation of the trained MERMAIDE intervention policy, showing that it outperforms one-shot adapted baselines on unseen test agents. We further include a "No Intervention" baseline to show how agents behave by default. In all, our results show that MERMAIDE's model-based meta-learning approach is highly effective: the principal obtains a higher score across agents with different learning algorithms and explore-exploit behaviors.

**Out-of-distribution performance.** Table 1 shows the principal's score when evaluated on test agents with the same algorithm but a *different exploration constant* than train agents. Using meta-learning (MF-MAML) and using a world model to predict the agent's behavior (WM-RL) both have advantages for training a robust and one-shot adaptable intervention policy. A world model is advantageous when 1) the test agent is more exploratory than the train set (e.g., $\epsilon = 0.1$ at training, $\epsilon = 0.4$ at test), or 2) the agent explores throughout an episode and is likely to often select actions other than the one with its current maximum mean reward estimate (e.g., $\epsilon = 0.5$ at training). Because we evaluate on $K = 1$, fine-tuning on only a single test-time episode, a trained world model provides a useful prior belief for the principal. Indeed, the MF-RL results show the hidden state representation of the model-free principal might be unable to adapt to high environment non-stationarity without a trained next-agent-action world model.

Compared to an $\epsilon$-greedy agent, the UCB agent explores mostly at the start of an episode, for all $\beta$. Hence, with UCB agents, the principal learns an effective one-shot adaptable intervention policy using meta-learning (MF-MAML) only (even without a world model), as the agents cause less distribution shift across different $\beta$. It further emphasizes the effectiveness of meta-learning for adaptive policy learning: unlike MF-MAML, neither the world model nor the intervention policy is meta-learned in WM-RL. Moreover, it also shows that for the same amount of distribution shift (characterized in Appendix B.2), the relative benefit of a world model or meta-learning depends on the nature of the agent's exploration strategy (which is unknown to the principal).

Table 7, Appendix C includes observations from training with additional random seeds for the experiments with MERMAIDE ($K = 1$) in Table 1.

**Agent exploration vs intervention cost.** In order to intervene effectively, the principal should learn *when* to intervene and *how much* to incentivize the agent while minimizing its incurred cost. This is a challenging learning problem for the principal not just during training, but more so during one-shot adaptation at test time. Bandit algorithms like EXP3 (Auer et al., 2002) use pessimism in the face of uncertainty, and encourage continued exploration. This increases the non-stationarity for the principal. In order to effectively incentivize such agents to prefer $a^*$, the principal needs to accurately predict the agent's policy from its observations; otherwise it can incur a high cost for intervening ineffectively *and* lowering its score, and learn to stop intervening. Indeed, our results when training on $\epsilon = 0.5$-greedy agents show that the MF-RL and MF-MAML principal stop intervening. In contrast, in that setting, MERMAIDE learns an effective intervention policy that outperforms all baselines, even under distribution shift between train and test agents.

**Cross-algorithm evaluation.** Table 2 shows the principal's scores in the zero-shot generalization setting when the training agent and test agents are of different types (different algorithms and different exploration coefficients). We consider only the WM-RL baseline to evaluate the generalization ability of the world model with standard policy gradients vs world model with meta-gradients, without adaptation to unseen test agents. We observe that when trained with UCB agents, MERMAIDE outperforms WM-RL for generalizing to $\epsilon$-greedy agents that have a lower exploration coefficient $\epsilon = 0.1$ or $0.2$. In contrast, when trained with $\epsilon$-greedy agents, MERMAIDE outperforms WM-RL for generalizing to UCB agents with both higher and lower levels of exploration. Note that the behavior of UCB agents is less stochastic than $\epsilon$-greedy agents. More generally, a meta-learning principal that is trained on a stochastic agent generalizes well to an equal or less stochastic agent in the zero-shot setting.

## 7 Limitations and Future Work

We have shown that MERMAIDE is an effective framework to learn principal intervention policies that adapt and generalize well to agents with unseen learning behavior. But our focus has been on stateless, sequential adaptive agents. Extending this setup to RL agents that solve non-Markovian settings (e.g., Markov Decision Processes) would introduce a more challenging learning problem for the principal and may require different neural network architectures for the principal's world model and intervention policy. Future work could also extend MERMAIDE to settings with multiple learning agents who may coordinate, compete, or a combination thereof. While our results show that MERMAIDE can adapt to bandit agents whose learning algorithm differs between training and test time, we would like to note that the meta-learning framework only guarantees adaptivity within the same meta-distribution over the training and test tasks. Finally, it would also be interesting to extend MERMAIDE to agents that adapt adversarially to the principal's intervention policy, which poses a challenging non-stationary problem for the principal.

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

Table 3: **Overview of notation.**

| Variable | Symbol |
|---|---|
| Time | $t$ |
| Principal | $p$ |
| Agent | $i$ |
| State | $s$ |
| State vector | $\boldsymbol{s}$ |
| State space | $S$ |
| Agent's action space | $A$ |
| Principal's action space | $A^p$ |
| Action sequence | $a_{1:T} = \{a_1, a_2, \ldots, a_T\}$ |
| Agent $i$'s reward sequence | $\tilde{r}_{1:T}^i = \{\tilde{r}_1^i, \ldots, \tilde{r}_T^i\}$ |
| Principal's reward sequence | $r_{1:T}^p = \{r_1^p, \ldots, r_T^p\}$ |
| Transition function | $\mathcal{P}$ |
| Agent $i$'s policy | $\pi^i$ |
| Principal's intervention policy | $\pi^p$ |
| Agent's mean estimate of intervened rewards for action $a$ | $\tilde{\mu}_a$ |
| Number of adaptation steps | $K$ |
| Number of meta-tasks for the planner | $N$ |
| Principal's history of interventions and observed agent actions upto time $t$ | $\mathcal{H}_t^p = \left\{a_1^p, a_1^i, a_2^p, a_2^i, \ldots, a_{t-1}^p, a_{t-1}^i\right\}$ |
| Agent's history of actions taken and rewards observed upto time $t$ | $\mathcal{H}_t^i = \left\{a_1^i, \tilde{r}_1^i, a_2^i, \tilde{r}_2^i, \ldots, a_{t-1}^i, \tilde{r}_{t-1}^i\right\}$ |

Table 4: **Notation for MERMAIDE** See Section Section 4 for their use.

| | |
|---|---|
| Principal's policy parameter | $\theta \in \Theta$ |
| Agent $i$'s learning algorithm | $f^i \in \mathcal{F}$ |
| Agent $i$'s true action mean rewards | $\mu^i \sim \mathcal{U}$ |
| Agent $i$'s intervened action mean rewards | $\tilde{\mu}^i$ |
| Principal's action at time $t$ | $a_t^p \sim \pi_\theta^p \left(a_t^p \mid a_{t-1}^i, a_{t-1}^p, \hat{a}_t^i, h_{t-1}^p\right)$ |
| Hidden state space of the principal's recurrent world model | $H$ |
| Agent's action at time $t$ | $a_t^i \sim \pi_t^i \left(a_t^i \mid \mathcal{H}_t^i\right), t = 1, \ldots, T$ |
| Agent's reward at time $t$ | $r_t^i \sim \mathcal{N}\left(\tilde{\mu}^i, \sigma^2\right)$ |
| Principal's world model estimate of the agent's action probability distribution | $\hat{\pi}_\omega^i : A \times A^p \times H \to \Delta\left(A\right), \ \hat{\pi}_{\omega,0}^i \in A$ |
| Principal's world model estimate of the latent state of the environment | $g_\omega^i : A \times A^p \times H \to H, \ g_{\omega,0}^i \in \Delta\left(H\right)$ |
| Principal's world model hidden state embeding in the LSTM architecture | $h_t^i = g_\omega^i \left(a_{t-1}^i, a_{t-1}^p, h_{t-1}\right), \ t = 2, \ldots, T \ h_1^i = g_{\omega,0}^i$ |

# A  Notation

For an overview of all symbols and variables used in this work, see Table 3 and Table 4.

# B  Additional Results

## B.1  Description of the bandit algorithms

We provide a brief overview of the learning algorithms referred to in Section 6.

**UCB.** This is an Upper Confidence Bound based exploration-exploitation algorithm that follows the principle of optimism in the face of uncertainty. At each time step $t$, the bandit agent selects an action

$$a_t = \arg\max_a \tilde{\mu}_a + \beta \sqrt{\frac{\log t}{n_a}} \qquad (5)$$

where $n_a$ is the number of steps until $t$ in which it previously selected the action $a$, $\tilde{\mu}_a$ is its corresponding mean estimate for the experienced rewards $\tilde{r}$ for action $a$ and $\beta$ is the exploration constant that balances the amount of exploration vs. exploitation across a time horizon $T$. A higher value of $\beta$ makes the agent less optimistic and explore its action space more. The UCB agent's tendency to explore is also affected by the difference in the mean reward estimates of its actions. In the context of our principal - agent problem formulation, if the UCB agent has a larger value of $\delta = \max_a r_a - r_{a^*}$, without any intervention at the beginning of an episode, its confidence bounds would quickly converge to exploiting the action $\arg\max_a r_a$. So a principal that intervenes only towards the later stages of an episode with this agent would have to provide much more incentives (higher $r'$) to alter the agent's preferred action to be $a^*$, thus incurring a larger cost $c$ as compared to a principal that intervenes more at the beginning of an episode when the UCB agent is still exploring its action space. This is also illustrated in Section 5 with a simpler best response agent in the single round game setting. As shown in Figure 2b and Figure 2c, under observation noise (partial information), the meta-trained principal has a better one-shot meta-test-time performance when the agent's base payoff has a higher difference between the principal's preferred action and the agent's intrinsic preference without any intervention. Additionally, Appendix B.3 provides an illustration of this behavior.

**$\epsilon$-greedy.** A simple exploration-exploitation strategy in the bandit setting is the $\epsilon$-greedy rule (Sutton & Barto, 1998) wherein the agent selects with probability $1-\epsilon$ the action $a_t = \arg\max_a \tilde{\mu}_a$ and with probability $\epsilon$ it selects a random action. In our setting, we consider $\epsilon$ to be constant during an episode, which results in a uniform exploration rate throughout. In contrast to the UCB agent, the $\epsilon$-greedy algorithm simulates a less optimistic, more exploratory agent for which the principal requires a robust belief representation of the agent's predicted behavior conditioned on the principal's past observations (Table 1). Since there is a uniform exploration rate for the agent, the principal has to continue intervening intermittently throughout an episode, especially when $\delta$ is large and the agent could obtain a higher reward for an action $a \neq a^*$ by exploring its action space when the principal does not intervene.

**EXP3.** The Exponential-weight algorithm for Exploration and Exploitation (EXP3) (Auer et al., 2002) follows a more pessimistic approach to exploration-exploitation in the bandit setting. It maintains a set of weights for each agent action $a \in A$ which are updated using the experienced rewards $\tilde{r}$ as follows:

$$\pi_t(a_t) = \frac{w}{|A|} + (1-w) \frac{\eta \exp\left(S_{a_t,t}\right)}{\sum_{a_t \in |A|} \eta \exp\left(S_{a_t,t}\right)}, \qquad (6)$$

where

$$S_{a_t,t} = \sum_{l=1}^{t} \mathbf{1}\left\{a_l = a_t\right\} \frac{\tilde{r}_{a_t,l}}{\pi_l}, \; \eta = \frac{w}{|A|}. \qquad (7)$$

Here, $w$ is the variable that determines the extent of uniform random exploration in the action space. This presents a very challenging problem to learn a suitable belief representation for such agents that can be utilized by a principal to guide its intervention policy. In Section 6, we exclude EXP3 from Table 1 since it is primarily designed for an adversarial bandit setup, whereas we do not consider an agent to have such biases under our current problem formulation.

## B.2 Characterizing the distribution shift in our evaluation setup

Bandit agents having the same base reward $r$ make different explore-exploit decisions depending on their algorithm (eg. UCB, $\epsilon$-greedy) and also their prior observations. In Section 6, we consider agents with the same set of base rewards, but following different bandit algorithms. Both UCB and $\epsilon$-greedy have tunable

Table 5: **Experiment design choice.** Frequency of agent selecting $a_t \neq \arg\max_a r_a$ with UCB and $\epsilon$-greedy algorithms on the same set of base rewards (without any intervention) with a horizon $T = 200$, averaged across 3 random seeds.

| $\beta$ | UCB | $\epsilon$-greedy | $\epsilon$ |
|---|---|---|---|
| 0.17 | 33 (0) | 33 (0) | 0.10 |
| 0.27 | 47 (0) | 47 (4) | 0.20 |
| 0.42 | 70 (0) | 68 (9) | 0.30 |
| 0.50 | 80 (0) | 81 (3) | 0.40 |
| 0.67 | 99 (0) | 99 (1) | 0.50 |

parameters that determine their explore-exploit tradeoff. In order to measure the robustness of the learnt principal policy to different agent behavior (leading to different levels of non-stationarity in the principal's environment between training and test agents), we vary the amount of exploration performed by the agent by varying the respective parameters: $\beta$ for the UCB agent and $\epsilon$ for the $\epsilon$-greedy agent. Table 5 shows the average (and standard error) frequency of exploration by the agents for our choices of $\beta$ and $\epsilon$ in Section 6. We vary $\beta$ and $\epsilon$ such that they are pairwise comparable in Table 1 and would lead to similar change in exploration frequency for both UCB and $\epsilon$-greedy agents. In other words, following Table 1, a principal trained with UCB agents having $\beta = 0.17$ when evaluated with UCB agents having $\beta \in \{0.17, 0.27, 0.42, 0.50, 0.67\}$ will encounter a similar shift in the agent's exploration frequency as in the case of training with $\epsilon$-greedy agents with $\epsilon = 0.1$ and evaluating on $\epsilon$-greedy agents having $\epsilon \in \{0.1, 0.2, 0.3, 0.4, 0.5\}$. In that case, the difference in achieved scores between the UCB and $\epsilon$-greedy agents can be attributed to the way in which they distribute their exploratory actions: UCB agent being more optimistic focuses most of its exploration at the beginning of an episode, whereas the $\epsilon$-greedy agent is more stochastic with uniform random exploration throughout.

### B.3 Visualizing the effect of $\beta$ and the effect of different types of principal's interventions on the behavior of a UCB agent

We will consider three different instances of base rewards for a UCB agent and characterize its behavior when

- unintervened

- the principal intervenes once every 10 time steps for $T = 200$ (strategy S1)

- the principal intervenes continuously until the first 20 time steps for $T = 200$ (strategy S2)

for $\beta \in \{0.17, 0.27, 0.42, 0.50, 0.67\}$. Instead of a learned stochastic intervention policy, we will analyze the effect of the deterministic policies S1 and S2 in aligning the preferred action of an agent with the preferred action of the principal. Note that both S1 and S2 incur the same total intervention cost.

First, consider a UCB agent whose base reward is $\boldsymbol{r} = [0.16, 0.11, 0.66, 0.14, 0.20, 0.37, \boldsymbol{0.82}, 0.10, \boldsymbol{0.84}, 0.10]$ (Figure 4). The principal prefers the action with base reward 0.82, whereas the unintervened agent would prefer the action with base reward 0.84. For such small value of $\delta = \max_{a \in A} \boldsymbol{r}^i[a] - \boldsymbol{r}^i[a^*] = 0.02$, the agent can be incentivized to align its preferred action with $a^*$ more easily than if $\delta$ were larger. Figure 4 depicts the behavior of the UCB agents with different exploration coefficients $\beta$ under different principal-agent interaction conditions over $T = 200$ as follows. In Figure 4a, a value of 1 indicates the time step when the unintervened agent selects the action with base reward 0.84, whereas Figure 4b similarly shows its frequency of selecting $a^*$ without the principal's intervention. As expected, these frequency distributions are quite similar since $\delta = 0.02$. Given the same budget for interventions, Figure 4c shows the agent's response to the principal's intervention strategy S1 and Figure 4d shows its response to the intervention strategy S2. In these figures, a value of 1 indicates the time step when the agent selects $a^*$. Both the principal intervention policies are able to align the action preference of the UCB agent with the principal's preference. But S2 receives a higher score, especially for agents with lower values for $\beta$. This is because the UCB algorithm gradually shifts from

exploration to exploitation as the episode progresses, and since S2 uses its intervention budget in the initial 20 time steps of the episode, the principal is able to incentivize the agent by effectively changing the observed reward $\tilde{r}$ at the beginning of the episode. It experimentally verifies our discussion in Section 6: in sequential learners, the time step *when* the principal intervenes determines how effective the intervention will be in helping align the agent's action preference with that of the principal. This becomes more pronounced for higher values of $\delta$, which we analyze next.

Figure 5 demonstrates the behavior of a UCB agent with base reward $\boldsymbol{r} = [0.32, 0.67, 0.13, 0.72, 0.29, 0.18, \boldsymbol{0.59}, 0.02, \boldsymbol{0.83}, 0.01]$. The principal prefers the action with base reward 0.59. Without any interventions, the agent prefers the action with base reward 0.83. In this case, $\delta = \max_{a \in A} \boldsymbol{r}^i[a] - \boldsymbol{r}^i[a^*] = 0.24$. Figure 5b indicates the frequency with which the unintervened agent selects $a^*$ whereas Figure 5a shows the frequency of selecting the action with base reward 0.83 without the principal's intervention. Note that the unintervened agent would rarely pick $a^*$, even less so for smaller values of the exploration coefficient $\beta$. Figure 5c indicates the frequency of the agent selecting $a^*$ with principal's intervention policy S1 and Figure 5d indicates its frequency of selection $a^*$ with principal's intervention policy S2. We observe that S2 outperforms S1 in aligning the agent's preferred action with the principal's preferred action. Since UCB agents tend to explore their action space more at the beginning of the episode, intervening on the agent's experienced reward during the initial time steps (S2) has a more noticeable effect in influencing the agent's preferred action than intervening with a fixed interval (S1).

Similar observations also hold in Figure 6 where the UCB agent has a base reward $\boldsymbol{r} = [0.79, 0.53, 0.57, \boldsymbol{0.93}, 0.07, 0.09, \boldsymbol{0.02}, 0.83, 0.78, 0.87]$ and the principal prefers the action with base reward 0.02. In this case, $\delta = \max_{a \in A} \boldsymbol{r}^i[a] - \boldsymbol{r}^i[a^*] = 0.91$. As Figure 6b shows, this implies that the unintervened agent would almost never select $a^*$ even when it has a higher exploration coefficient $\beta$. Even with intervention policy S1, over $T = 200$, the principal wouldn't be able to align the agent's preferred action with its own as shown in Figure 6c. In contrast, Figure 6d shows that a principal with intervention policy S2 would outperform S1 and achieve a higher score, but the agent eventually discovers its own preference when the intervention stops in S2 and then it no longer selects $a^*$. This further highlights the extent of non-stationarity in the environment that affects the intervention policy of the principal and also the learning behavior of the agent. It also demonstrates the difficulty of the learning problem in our setup and the importance of learning a cost-efficient few-shot adaptable principal intervention policy to effectively intervene on unknown adaptive agents.

### B.4   Description of baselines

We now describe the details of our evaluated baselines in Section 6 along with their variations that assume access to an agent state oracle.

**Rule based intervention with an agent state oracle (RB):**   Given an oracle that correctly identifies the action $a_t$ to be taken by an agent in the next time step, a simple rule based approach is for the principal to intervene at time $t$ when $a_t \neq a^*$. We assume that the principal always intervenes with a fixed incentive ($r' = 0.5$ or $1$) and we compute the principal's maximum possible score. Note that this is not a realistic solution for the principal since it is impractical to expect the availability of such an oracle, especially for out of distribution test agents.

**Model-free learning based intervention policy:**   In this framework, we assume that the planner has a recurrent intervention policy that outputs a distribution over interventions $a_t^p \sim \pi_\theta^p \left( a_t^p | a_{t-1}^i, a_{t-1}^p, h_{t-1}^p \right)$, conditioned on the planner's intervention and observed agent action at $t - 1$. The policy network is trained using REINFORCE for the MF-RL baseline and using MAML for the MF-MAML baseline.

**Learned intervention policy with an agent state oracle:**   In this setting, the principal learns a recurrent intervention policy that outputs a distribution over interventions $a_t^p \sim \pi_\theta^p \left( a_t^p | a_t^i, h_{t-1}^p \right)$ conditioned on the true agent action at time $t$ provided by an oracle. The policy network is trained using REINFORCE for the SB-RL baseline and MAML for the SB-MAML baseline.

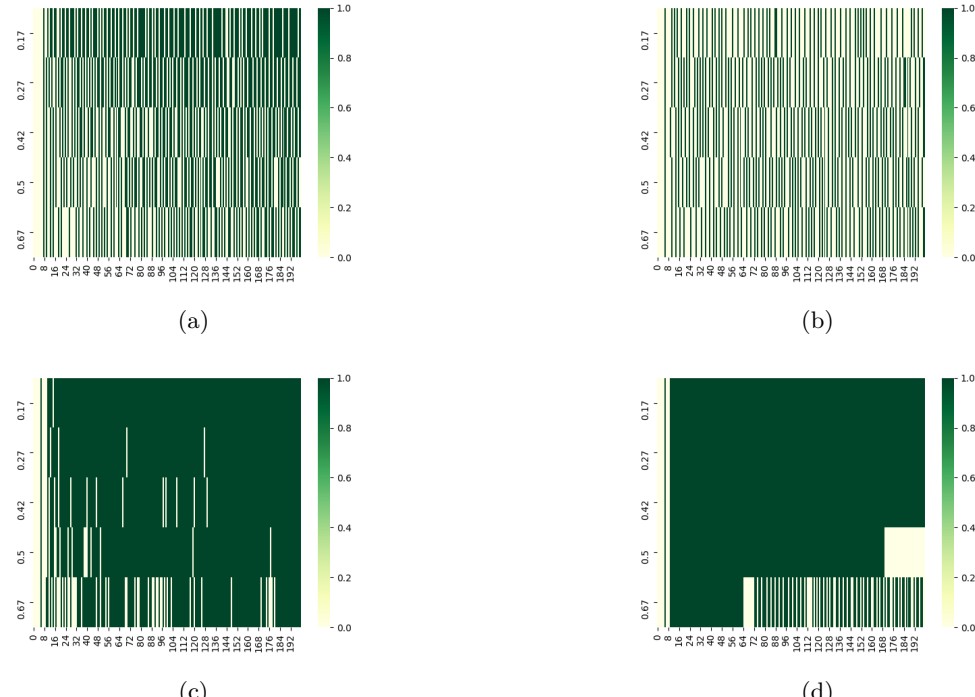

Figure 4: **Characterizing agent's behavior.** UCB agent with base rewards $[0.16, 0.11, 0.66, 0.14, 0.20, 0.37, \mathbf{0.82}, 0.10, \mathbf{0.84}, 0.10]$. The agent prefers the action with base reward 0.84, while the principal prefers the action with base reward 0.82. Horizontal axis indicates time steps $t = \{1, \ldots, 200\}$. Vertical axis indicates agents following UCB with different exploration coefficient $\beta$. Values are either 0 or 1. (a) Frequency distribution of agent selecting its unintervened preferred action with base reward 0.84. (b) Frequency distribution of agent selecting $a^*$ without principal's intervention. (c) Frequency distribution of agent selecting $a^*$ under principal's intervention S1. (d) Frequency distribution of agent selecting $a^*$ under principal's intervention S2. For a small $\delta = \max_{a \in A} \mathbf{r}^i[a] - \mathbf{r}^i[a^*] = 0.02$, both S1 and S2 affect the agent's behavior quite similarly.

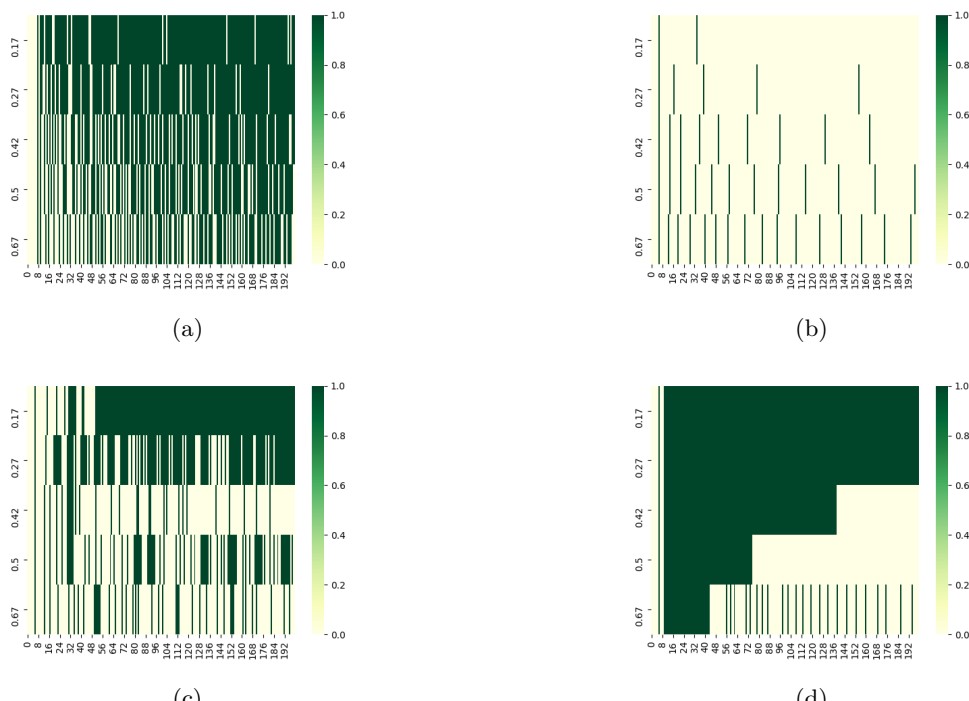

(a)

(b)

(c)

(d)

Figure 5: **Characterizing agent's behavior.** UCB agent with base rewards $[0.32, 0.67, 0.13, 0.72, 0.29, 0.18, \textbf{0.59}, 0.02, \textbf{0.83}, 0.01]$. The agent prefers the action with base reward 0.83, while the principal prefers the action with base reward 0.59. Horizontal axis indicates time steps $t = \{1, \ldots, 200\}$. Vertical axis indicates agents following UCB with different exploration coefficient $\beta$. Values are either 0 or 1. (a) Frequency distribution of agent selecting its unintervened preferred action with base reward 0.83. (b) Frequency distribution of agent selecting $a^*$ without principal's intervention. (c) Frequency distribution of agent selecting $a^*$ under principal's intervention S1. (d) Frequency distribution of agent selecting $a^*$ under principal's intervention S2. For different values of $\beta$, the UCB agent acts differently based on *when* the principal intervened following S1 or S2. S1 intervenes periodically whereas S2 intervenes only at the beginning. The action selected by the agents in an episode clearly reflects the effect that this has in being able to align the agent's preference with that of the principal.

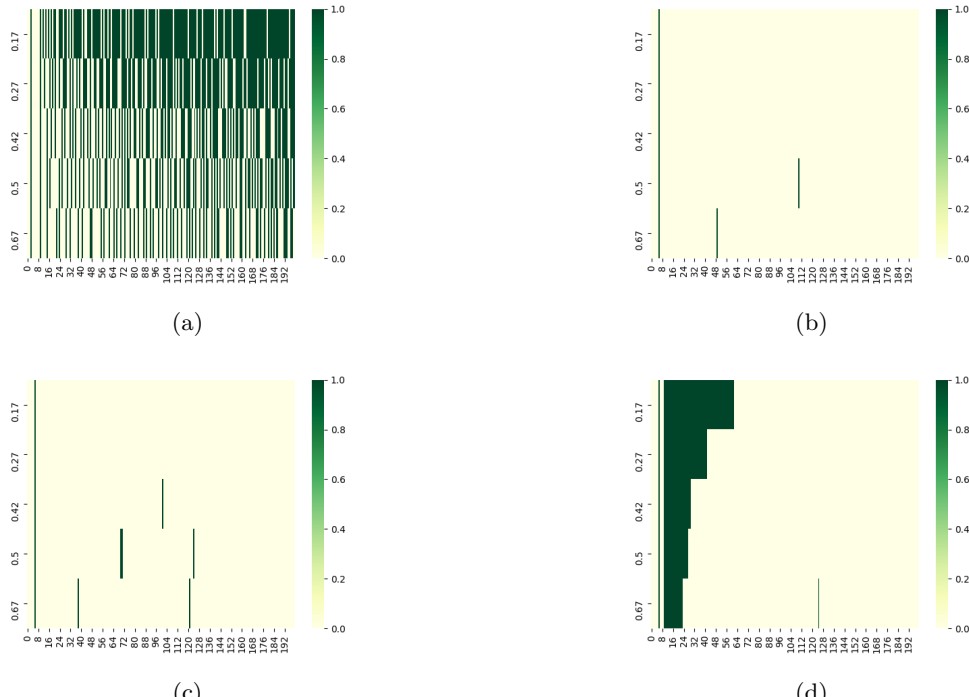

Figure 6: **Characterizing agent's behavior.** UCB agent with base rewards $[0.79, 0.53, 0.57, \mathbf{0.93}, 0.07, 0.09, \mathbf{0.02}, 0.83, 0.78, 0.87]$. The agent prefers the action with base reward 0.93, while the principal prefers the action with base reward 0.02. Horizontal axis indicates time steps $t = \{1, \ldots, 200\}$. Vertical axis indicates agents following UCB with different exploration coefficient $\beta$. Values are either 0 or 1. (a) Frequency distribution of agent selecting its unintervened preferred action with base reward 0.93. (b) Frequency distribution of agent selecting $a^*$ without principal's intervention. (c) Frequency distribution of agent selecting $a^*$ under principal's intervention S1. (d) Frequency distribution of agent selecting $a^*$ under principal's intervention S2. Since $\delta = \max_{a \in A} \boldsymbol{r}^i[a] - \boldsymbol{r}^i[a^*] = 0.91$ is large and the UCB agent explores its action space in the initial time steps, S2 is able to intervene more effectively than S1 and achieves a higher score. Since the agent is a sequential learner, it discovers its own preferred action once the principal stops intervening and does not select $a^*$.

Table 6: **Principal (with oracle agent state input) scores across 3 random seeds.** These baselines are not applicable in practice since they cheat by assuming access to an oracle that always informs them of the agent's next action. We include them here as a form of standardization with respect to a (perfect) system that does not face the challenges of partial observability or out-of-distribution generalization in our setting.

| Train on UCB, $\beta = 0.17$ | Test on $\beta = 0.17$ | $\beta = 0.27$ | $\beta = 0.42$ | $\beta = 0.5$ | $\beta = 0.67$ |
|---|---|---|---|---|---|
| *No intervention* | 3 (0) | 5 (0) | 8 (0) | 10 (0) | 12 (0) |
| RB | 173 (0) | 166 (0) | 154 (0) | 146 (0) | 126 (0) |
| SB-RL | 168 (3) | 138 (27) | 128 (26) | 122 (24) | 107 (22) |
| SB-MAML | 169 (3) | 169 (1) | 155 (2) | 148 (1) | 128 (2) |
| **Train on $\epsilon$-greedy, $\epsilon = 0.1$** | $\epsilon = 0.1$ | $\epsilon = 0.2$ | $\epsilon = 0.3$ | $\epsilon = 0.4$ | $\epsilon = 0.5$ |
| *No intervention* | 3 (0) | 4 (1) | 7 (0) | 9 (1) | 11 (0) |
| RB | 156 (3) | 130 (1) | 105 (4) | 87 (4) | 62 (6) |
| SB-RL | 148 (2) | 119 (3) | 87 (4) | 75 (6) | 50 (2) |
| SB-MAML | 152 (1) | 126 (2) | 105 (3) | 66 (3) | 30 (9) |
| **Train on UCB, $\beta = 0.67$** | $\beta = 0.17$ | $\beta = 0.27$ | $\beta = 0.42$ | $\beta = 0.5$ | $\beta = 0.67$ |
| *No intervention* | 3 (0) | 5 (0) | 8 (0) | 10 (0) | 12 (0) |
| RB | 173 (0) | 166 (0) | 154 (0) | 146 (0) | 126 (0) |
| SB-RL | 166 (3) | 163 (2) | 150 (3) | 146 (2) | 128 (2) |
| SB-MAML | 173 (1) | 170 (0) | 159 (0) | 152 (0) | 133 (0) |
| **Train on $\epsilon$-greedy, $\epsilon = 0.5$** | $\epsilon = 0.1$ | $\epsilon = 0.2$ | $\epsilon = 0.3$ | $\epsilon = 0.4$ | $\epsilon = 0.5$ |
| *No intervention* | 3 (0) | 4 (1) | 7 (0) | 9 (1) | 11 (0) |
| RB | 156 (3) | 130 (1) | 105 (4) | 87 (4) | 62 (6) |
| SB-RL | 49 (46) | 51 (35) | 64 (29) | 61 (15) | 28 (17) |
| SB-MAML | 93 (45) | 62 (32) | 32 (13) | 58 (25) | 24 (17) |

**Learned intervention policy with a world model without meta-learning (WM-RL):** In this setting, we use our proposed recurrent world model with a recurrent intervention policy trained using REINFORCE. Here, the policy network outputs a distribution over interventions $a_t^p \sim \pi_\theta^p \left( a_t^p | \hat{a}_t^i, h_{t-1}^p \right)$ where $\hat{a}_t^i = \arg\max_a \hat{\pi}_\omega \left( a_t^i | a_{t-1}^i, a_{t-1}^p, h_{t-1}^i \right)$.

We would like to highlight an implementation detail in our baselines indicated 'RL' in Section 6. Since we evaluate our learnt principal policy in the $K$-shot adaptation setting which is common in the meta-learning literature, we ensure that the principal policies that are not meta-trained are also allowed to $K$-shot adapt at test time. This means that the 'RL' policies are also updated at test time, before evaluation, using $K$ rounds of principal-agent interactions. This is in contrast to Section 5 where 'RL' was trained from scratch during test time adaptation. It further shows that even with pre-training (on the same set of train agents as used by 'MAML'), standard policy gradient update does not lead to effective test time $K$-shot adaptation on test agents.

In Table 6, we compare the test time scores for the principal policy having access to a state based oracle. We observe that overall, the meta-trained principal policy (SB-MAML) achieves a higher score even with distribution shift across different bandit algorithms and different levels of exploration, compared to the SB-RL baseline. The rule based baseline also shows strong performance but we note its scores do not reflect adaptation to distribution shift. However, none of these baselines that assume the principal has access to an oracle that correctly predicts the agent's action at the next time step are realistic. We can only treat the scores in Table 6 as gold standards in a perfect system that does not account for the challenges faced by a principal in practice.

**Training details.** In Section 5, the principal policy $\pi^p$ is a fully connected neural network (MLP) with one hidden layer and ReLU activation. Given an (noisy) observed value of the agent type as input, it predicts the probability of intervention: $\pi_t^p$. The principal's action at time $t$ is $a_t^p \sim \text{Bern}\left(\pi_t^p\right)$.

---

**Algorithm 2** MERMAIDE ($K$-shot Adaptation)

---

1: Initialize principal with trained parameters $(\theta_{\text{meta}}, \omega_{\text{train}})$, and hidden states $h_0^i, h_0^p$.
2: **for** agents (tasks) $i = 1, \ldots, n_{\text{test}}$ **do**
3:   Initialize agent: $(\mu^i, \pi_0^i)$, task specific intervention policy parameter $\theta\left(\tau_0^i\right) = \theta_{\text{meta}}$.
4:   **for** $k = 1, \ldots, K$ **do**
5:     **for** time t $= 1, \ldots, T$ **do**
6:       Predict $\hat{a}_t^i = \arg\max_{a_t^i} \hat{\pi}_{\omega_{\text{train}}}\left(a_t^i | a_{t-1}^i, a_{t-1}^p, h_{t-1}^i\right)$ using the world model.
7:       Intervention: $\tilde{\mu}^i = \mu^i + a_t^p, \quad a_t^p \sim \pi_{\theta\left(\tau_k^i\right)}^p\left(a_t^p | a_{t-1}^i, a_{t-1}^p, \hat{a}_t^i, h_{t-1}^p\right)$.
8:       Agent acts: $a_t^i \sim \pi_t^i$ and receives reward $r_t^i \sim \mathcal{N}\left(\tilde{\mu}^i, \sigma^2\right)$. $\pi_t^i \mapsto \pi_{t+1}^i$.
9:     **end for**
10:     Locally update $\theta\left(\tau_k^i\right) \mapsto \theta\left(\tau_{k+1}^i\right)$. {Using REINFORCE.}
11:   **end for**
12:   **for** $t = 1, \ldots, T$ **do**
13:     Predict $\hat{a}_t^i = \arg\max_{a_t^i} \hat{\pi}_{\omega_{\text{train}}}\left(a_t^i | a_{t-1}^i, a_{t-1}^p, h_{t-1}^i\right)$ using the world model.
14:     Intervention: $\tilde{\mu}^i = \mu^i + a_t^p, \quad a_t^p \sim \pi_{\theta\left(\tau_K^i\right)}^p\left(a_t^p | a_{t-1}^i, a_{t-1}^p, \hat{a}_t^i, h_{t-1}^p\right)$.
15:     Agent acts: $a_t^i \sim \pi_t^i$, receives reward $r_t^i \sim \mathcal{N}\left(\tilde{\mu}^i, \sigma^2\right)$. Updates $\pi_t^i \mapsto \pi_{t+1}^i$.
16:     Update principal's score.
17:   **end for**
18: **end for**

---

For the 'RL' principal, it is trained on the test agents starting from scratch over $K$ episodes before evaluation. For the MAML principal, it is meta-trained to learn an initial parameterization with a different set of training agents and evaluated with $K$-shot adaptation on the test agents.

In Section 6, the recurrent world model and policy networks are GRUs with 2 layers and hidden state dimension 128. For meta-training, the inner gradient update loop uses SGD optimizer with a learning rate of $7 \times 10^{-4}$ whereas the meta-update step uses Adam with a learning rate of 0.001. The world model is trained only with the set of training agents, it is not adapted at test time: only the policy network is $K$-shot adapted.

We plan to release the code for our implementation with the published paper.

### B.5   Overview of $K$-shot adaptation with MERMAIDE:

Algorithm 2 outlines our framework for $K$-shot adaptation of the meta-trained principal to test agents. In our experiments, $K = 1$.

## C   Additional experimental results with bandit agents

**Comparing the stability of MERMAIDE for different sets of random seeds.**   We trained and evaluated MERMAIDE for 3 additional random seeds in the $K = 1$-shot setting from Table 1. In Table 7, we show the scores (mean and standard error) for both sets of seeds and all the six seeds combined. Our results indicate that the training and evaluation of MERMAIDE is stable, with reasonable and explainable variability across random seeds. More specifically, models trained with different sets of random seeds result in similar mean scores with small standard error when $K = 1$-shot evaluated on the less stochastic UCB agents. In contrast, evaluation with the more stochastic $\epsilon$-greedy agents results in comparably larger standard error and more variation in the mean scores for models trained with different sets of random seeds. Due to the computational costs involved, we were unable to evaluate all baselines with six seeds but on the basis of these observations, we do not expect a significant deviation from the claims made in Section 6 and the values originally reported in Table 1 and Table 2, even with more seeds.

Table 7: **Comparing the variability of MERMAIDE test scores with a trained model across different sets of random seeds.** Set 1 uses the seeds {11, 26, 90} and Set 2 uses the seeds {12, 27, 91}. 'Combined' indicates mean and s.e. scores for models trained with the seeds {11, 12, 26, 27, 90, 91}. Overall, we observe that training MERMAIDE with different sets of random seeds shows little variance in the evaluation scores within error bounds, especially with the less stochastic UCB agent. For the $\epsilon$-greedy agent, we observe a higher standard error in the $K = 1$-shot evaluation scores, but it is in line with the stochasticity associated with $\epsilon$-greedy action selection in the bandit agent.

| **Train on UCB,** $\beta = 0.17$ | Test on $\beta = 0.17$ | $\beta = 0.27$ | $\beta = 0.42$ | $\beta = 0.5$ | $\beta = 0.67$ |
|---|---|---|---|---|---|
| MERMAIDE (Set 1; 3 seeds) | 154 (2) | 151 (1) | 129 (1) | 129 (1) | 87 (0) |
| MERMAIDE (Set 2; 3 seeds) | 144 (3) | 133 (3) | 122 (2) | 108 (2) | 81 (2) |
| MERMAIDE (Combined; 6 seeds) | 151 (3) | 142 (4) | 125 (2) | 118 (5) | 84 (2) |
| **Train on UCB,** $\beta = 0.67$ | $\beta = 0.17$ | $\beta = 0.27$ | $\beta = 0.42$ | $\beta = 0.5$ | $\beta = 0.67$ |
| MERMAIDE (Set 1; 3 seeds) | 132 (1) | 130 (1) | 123 (2) | 115 (2) | 99 (1) |
| MERMAIDE (Set 2; 3 seeds) | 119 (7) | 118 (5) | 110 (3) | 104 (3) | 87 (2) |
| MERMAIDE (Combined; 6 seeds) | 126 (4) | 124 (3) | 116 (3) | 110 (3) | 93 (3) |
| **Train on** $\epsilon$**-greedy,** $\epsilon = 0.1$ | $\epsilon = 0.1$ | $\epsilon = 0.2$ | $\epsilon = 0.3$ | $\epsilon = 0.4$ | $\epsilon = 0.5$ |
| MERMAIDE (Set 1; 3 seeds) | 138 (1) | 112 (2) | 85 (3) | 66 (2) | 37 (4) |
| MERMAIDE (Set 2; 3 seeds) | 133 (2) | 133 (4) | 133 (4) | 133 (4) | 133 (4) |
| MERMAIDE (Combined; 6 seeds) | 136 (2) | 123 (5) | 109 (11) | 99 (15) | 85 (22) |

