# OpenReview forum: "MERMAIDE: Learning to Align Learners using Model-Based Meta-Learning"
_TMLR — Accepted by TMLR_

### Review · Reviewer_7t2x · 2023-07-18

**Summary Of Contributions:**

**Summary:**
The paper investigates how to learn how to intervene on a learner’s reward function such that the learner takes a desired action with high probability, when (1) such interventions are costly and (2) the learner’s type (precise reward function, exploration strategy, and policy update function) are unknown and may be different between train- and test-time. The proposed solution is to meta-train a principal over many learner types such that it generalizes to learners from the same distribution at test time, and empirically performs well on learners from different distributions. The approach is split into two parts (learned simultaneously): learning a (world-) model over how a reward-intervention in a particular time-step affects the learners’ highest-probability action; and learning an optimal policy over interventions (by using the world-model) that takes into account the cost of intervening. Importantly, both parts do not explicitly observe the learners type, but through meta-training over many learner types behave as if inferring the learner’s type on the fly and adapting accordingly (amortized inference). Overall this leads to a partially observable, non-stationary learning problem for both the principal and the learners. The proposed approach is empirically validated in a series of experiments on a simple one-shot strategic game (Stackelberg game), with ablations regarding partial observability and non-stationarity. A second set of experiments is shown on a bandit task, where out-of-distribution performance of the principal is also evaluated. On both types of experiments the proposed algorithm performs well compared to baseline methods and alternative approaches.

**Contributions:**
 1. Formulation of incentive/reward design under unknown learner (-type) and intervention-cost as a meta-learning problem. The formulation is sensible and quite general (ranging from fully observable one-shot settings to partially observable non-stationary problems). While this may not be the very first formulation as a meta-learning problem, the paper executes it well, provides good motivation and a nice introduction / discussion of design choices.
 2. Proposal of novel algorithm (MERMAIDE) using a meta-trained principal (gradient-based meta-learning and RL with a learned world-model). A sound proposal, with good discussion of (most) design choices. The algorithm itself is applicable to other coupled cooperative/competitive problems with two interacting learners (one doing first order strategic optimization, the other one doing second-order strategic optimization), making it relevant beyond alignment/incentive design.
 3. Empirical evaluation. Well executed but somewhat limited in scope. A big plus is that there are a number of ablations and out-of-distribution evaluations. One small point of criticism, given the intro, is that all results are simulation only (the advertised sim-to-real advantage of the method is not evaluated).


**Audience:**

Yes

**Broader Impact Concerns:**

No significant broader impact concerns.

As with any type of incentive design, there is potential to abuse the technology to 'nudge' humans towards actions that maximize economic profit or benefit a political agenda. I think a brief broader impact statement would be good, but would not object to the paper's publication if there is none (since I'd consider the paper quite far removed from an actual application for human/user incentive design).

**Claims And Evidence:**

Yes

**Requested Changes:**

**Major (please address):**
 1. Include a dedicated limitations paragraph towards the end of the paper, discussing to which extent the broader motivation at the beginning of the paper has been addressed and what the most important shortcomings of the current algorithm and experiments are.

**Minor (suggestions, typos, etc. - not crucial for publication):**

 2. Strictly speaking, meta-leraning only guarantees generalization at test time within the meta-distribution. In practice it has of course been observed many times that meta-trained systems generalize (sometimes considerably) beyond the meta-distribution. Please mention this.
 3. “In this work, we focus on agents in a stateless environment for ease of exposition” - fine, but it may also make the learning problem for the principal significantly harder if the environments were stateful (the current phrasing sounds a bit like it would just make the math. notation more complex). Please mention this in the limitations paragraph.
 4. (Similar to 3.) P4:  “These forms of distribution shift distinguish our adaptive intervention policy learning setting from most prior work in meta-learning, which often assume stationarity within a task and also assume similar task distributions at train and test times” At least implicitly (since meta-learning does the heavy lifting) MERMAIDE also somewhat relies on the second assumption (similar task distribution at train and test times) - whether or not the meta-trained principal generalizes to a new task distribution is not theoretically guaranteed but can only be empirically determined.

 5. P5: MERMAIDE step 2: why only use the most likely predicted action rather than the full distribution over next actions provided by the world model? Could it not make a difference to the principal’s policy if the principal’s preferred action is the second most likely or the least likely action according to the world model?

 6. Algorithm 1, line 10 and 17: Why does the agent’s action not appear in the Gaussian reward. I would have expected something like a negative squared error between $a_t^i$ and $\tilde{\mu^i}$ with added Gaussian noise: $r_t^i \sim \mathcal{N}(-(a_t^i - \tilde{\mu^i})^2, \sigma^2)$.

 7. P2 last paragraph: sentence starts with ‘However’ - a bit unclear what that refers to (is something missing?).

 8. P7 typo: “using MAML than RL”

 9. P7: At test-time (for the Stackelberg experiments) what is the distribution that agent types are drawn from? The same as the meta-training distribution?

 10. Fig 3a: what does the colorbar indicate (label/description in caption missing; though given in main text)?

 11. Fig 3 b,c: Is the number of interventions normalized by episode length (why does it lie between 0 and 1)?

 12. P10: “Results.”: How were the base rewards for the train- and test-time agents constructed?

 13. Table 2: Why the two empty entries (last column, row 3 and 4)?


**Strengths And Weaknesses:**

**Pro:**
 1. Very well written paper, particularly the introduction and proposal of the method. The experiments section gets a bit dense at times, but still at a high-quality level of writing.
 2. Two very clearly spelled out claims that are tested. While the intro refers to a broader set of potential claims (which is good to situate the work), the “hard” take-aways are clearly spelled out.
 3. Empirical evaluation includes important ablations and out-of-distribution experiments.

**Con:**
 1. No explicit discussion of limitations of the current study (also w.r.t. the broader goals alluded to in the intro). Would be great to see a paragraph towards the end of the paper.
 2. One promise of the method is that it lends itself well to training a principal via imprecise simulations which then allows for rapid transfer to real-world incentive/reward design. The OOD experiments certainly support that claim but a user-study with humans would make such a claim much stronger. I do not consider this necessary for publication, but if included it would make the paper even stronger.
 3. Going from a bandit setting to a full RL setting is likely to make the learning problems for the principal quite a bit harder. While results on the bandits look promising; showing results in some RL setting (e.g. gridworlds) would further strengthen the paper. Similar to 2., I do not consider this necessary for publication.

**Verdict:**
Overall I think the main idea of the paper is technically sound, well executed, and empirical results clearly support the two main claims (and the method more generally). The paper does a good job motivating the particular algorithm that is proposed. Perhaps the two biggest improvements for making the paper even stronger are on the experimental side: showing that the learning problem for the principal is still easily solvable in a full RL environment (and whether/how the principal generalizes to learners in new environments); and an empirical study showing that the meta-trained principal transfers well to human learners. I do not consider either necessary for publication (just suggestions for improvements). Taking all of this together I think the paper is ready for publication. Looking forward to the other reviews and the authors’ feedback.

---

> ### Author Response · Authors · 2023-09-20
> **Response to review**
>
> We thank you for your time and thoughtful review. We address your requested changes as follows:
>
> 1. > a dedicated limitations paragraph
>
>     We have modified sec 7 in the updated paper to describe the limitations and future work.
>
> 2,3,4 :  We have addressed these concerns by appropriately updating the paper.
>
> 5. > why only use the most likely predicted action rather than the full distribution over next actions provided by the world model?
>
>     It might be an interesting aspect of the problem to consider, if we changed our principal’s action space to intervene by different amounts on the agent’s different actions. For example, in the current action space definition, the principal can intervene by adding +0.5 (or +1) to its preferred action and -0.5 (or -1) to all other actions in the bandit’s action space. But if the principal could vary the amount of positive or negative incentives it applies for the different actions in the agent's action space, it would be a harder decision making problem and may likely benefit more by considering the distribution over the world model’s predicted actions. However, we note that the principal considering the entire distribution over next actions might not scale to larger agent action spaces.
>
> 6. > Algorithm 1, line 10 and 17
>
>     Considering the bandit setting as an example, $\tilde{\mu}^i$ refers to the  agent's experienced rewards $\tilde{\mathbf{r}}$ in eq 4, and so in algorithm 1, for action $a_t^i$, we can interpret the reward observed as $r_t^i[a_t^i] \sim \mathcal{N}(\tilde{\mu}^i[a_t^i], \sigma^2)$.
>
> 7,8 :  We have addressed these concerns by appropriately updating the paper.
>
> 9. > At test-time (for the Stackelberg experiments) what is the distribution that agent types are drawn from? The same as the meta-training distribution?
>
>     Yes, agents are sampled with $u \sim U(0,1)$.
>
>
> 10, 11 :  We have addressed these concerns by appropriately updating the paper.
>
> 12. > P10: “Results.”: How were the base rewards for the train- and test-time agents constructed?
>
>     They were sampled iid from $U(0,1)$. On page 8, we mention that "We assume $r_a \in (0, 1) \forall a$".
>
> 13. > Table 2: Why the two empty entries (last column, row 3 and 4)?
>
>     Unfortunately, we don't have access to these results at the moment.

---

> > ### Comment · Reviewer_7t2x · 2023-09-30
> > **Thank you for the detailed response and changes to the manuscript.**
> >
> > I consider all the issues raised in my review well addressed, and I have no further questions.

---

### Review · Reviewer_euPP · 2023-07-22

**Summary Of Contributions:**

This paper considers the setting of principal-agent problems where a principal may intervene at a cost on the payoffs given to agents for taking different actions, with the goal of incentivizing the agents to take a desired action. It proposes MERMAIDE, a deep reinforcement learning-based approach to learn a policy for the principal in this principal-agent problem, which leverages a combination of meta-learning and a world model on top of a deep RL algorithm. This approach is evaluated and ablated on both Stackelberg games and in a multi-armed bandit setting on a variety of agent policies.

**Audience:**

No

**Claims And Evidence:**

Yes

**Requested Changes:**

I have the following questions and requested changes:
- Provide examples of practical realizations of the problem setting where it would be beneficial to employ a deep RL algorithm.
- The authors should provide an explanation as to why they do not allow the principal’s policy to depend on the tilmestep t (or if I misinterpreted the paper, make it clearer that it does consider non-Markovian policies). I would also like to see a more rigorous discussion and analysis of the potential performance gap induced by this choice.
- While not necessary to include in revisions, at least in the author response I would like to see some justification for why an LSTM is used as the world model instead of a transformer which receives as context the entire history of the episode (and prior episodes in the few-shot setting). Since T is not too large in these settings, this should be computationally feasible and might allow for faster adaptation to the new agent via in-context learning.
- Provide some evidence that the vanilla RL agent’s poor performance is due to a limitation of RL in general, rather than being due to the instability of the particular algorithm used. For example, do the parameters converge to a fixed policy or do they diverge/oscillate? Does the loss go down?
- More generally, more discussion about *why* the methods that don’t use a world model or meta-learning tend to do worse would be useful to understand MERMAIDE.


**Strengths And Weaknesses:**

**Strengths**

  - The application of deep RL to mechanism design is not something I’ve seen attempted before. While deep RL has been used for a long time in multi-agent domains, historically these settings have not made assumptions on the opponent and action space that would correspond to principal-agent problems.

  - The approach uses simple, easy-to-understand algorithms (REINFORCE + MAML) that are widely regarded as the standard baselines in their respective fields, making it easy to interpret the results without ablations.

  - The method is evaluated under a set of tasks that incorporate distinct and interesting features, including the observability of the agent’s utility function, noise in the observations, the agent’s behavior policy, and the dimensionality of the agent’s action space.

  - The use of a world model (or perhaps more accurately a meta-imitation learner to infer the agent’s policy) in conjunction with meta-learning is a creative idea.

  - I appreciate the clear decomposition of the claims of the paper given in the introduction.

**Weaknesses**

  - The paper could be better motivated. The examples of principal-agent problems given in the introduction are not settings where I would ever expect employing a deep RL algorithm to be practical (e.g. government taxation policy has such a long period between steps that it would take many centuries before the RL agent would have meaningfully updated its parameters).

  - One of the main weaknesses of the paper is that the constraints on the principal’s policy to be markovian could significantly reduce the performance of the optimal policy in the principal’s hypothesis class. Presumably, in most cases the expected benefit of an intervention will depend on its effect on the agent’s propensity to cooperate in the future as well, and so interventions earlier in the trajectory will likely pay off more in terms of discounted return. It doesn’t look like the current formulation of the problem setting permits this.

  - Similarly, only using the agent’s next-step action to predict the efficacy of an action seems like it may not be the most effective use of a world model compared to evaluating the cumulative effect of an intervention on an agent’s later policies.

  - The only baselines compared against in the paper are other deep RL algorithms. Given that there has been a great deal of work on mechanism design in the game theory community, I would assume that (particularly for the stackelberg dynamics evaluation task) some online learning method could be used to learn a good intervention policy as well.

  -  For example, how does the deep RL approach perform compared to a principal which estimates the value of u based on prior experience and then follows an epsilon-greedy policy w.r.t. this estimate?

  - I would ideally like to also see some smarter online algorithms compared against as well. It is one thing to show that it is _possible_ to use deep RL in this setting, but I think it is also on the authors to show why it would be _desirable_.

  - The RL baseline using REINFORCE doesn’t seem to be doing any learning. It is possible that this is due to an innate failure of RL algorithms in this domain, but I could also believe, given the instability of REINFORCE, that the learner simply never converged.

---

> ### Author Response · Authors · 2023-09-20
> **Response to review**
>
> We thank you for your time and thoughtful review. We address your primary concerns as follows:
>
> 1. > Provide examples of practical realizations of the problem setting where it would be beneficial to employ a deep RL algorithm.
>
>     The principal-agent setting has applications in designing auctions, recommender systems, educational tutoring systems, and healthcare. Our approach using deep RL can be beneficial especially when a simulator is available for modeling different agent types and training the principal’s meta-learned policy. For example, there has been recent interest in developing RL-based methods for automated tutoring systems [1] and for automated testing [2]. In such scenarios, we could potentially frame the tutoring system as a mechanism design problem where the principal (tutor) is intervening (by designing tutoring lessons of appropriate difficulty) to align the agent(student)’s actions with that preferred by the principal. We have also briefly described two of these examples in the first para on page 1 of our updated paper.
>
> 2. > Dependence of the intervention policy on timestep t
>
>     In our setup, the agent is a learner that can update its policy based on its previous observed rewards, which also depend on the principal's intervention. This learning and adaptation process itself makes the principal's problem non-Markovian, and as we describe in page 10 (under "Challenges in the sequential setting"), the principal has to carefully decide _when_ and how much to intervene so that it can change the agent’s preferred action while incurring minimum cost. Further, we used a GRU to train the principal’s intervention policy. Prior work in deep RL has established that recurrent networks are capable of learning non-Markovian policies [3]. Therefore, in our proposed algorithm, the principal is not restricted to Markovian policies.
>
> 3. > Using transformers for the world model
>
>     We agree that transformers could be a potentially useful choice for training the world model, especially when the agents are stateful and equipped with more complex policies than the bandit agents. Recently, [4,5] showed the effectiveness of transformers for training a world model whereas previously, authors in [6] have demonstrated success with recurrent state space models. Our experiments show that in the principal-agent setup, even by using a simple model like the GRU, we were able to train effective intervention policies for the principal with bandit agents. That is, we do not need alternatives, e.g., transformers, to do good world modeling.
>
> 4. > Poor performance of vanilla RL
>
>     The poor performance of vanilla RL is because of the difference between the implementation of the RL and the MAML setup, which we briefly describe in page 21, para 2. At test time, the RL principal is allowed to train for K rounds (in the K-shot setup) and then evaluated, as a result for K=1 the RL policy is not able to learn the Stackelberg equilibrium intervention policy for the principal.
>
> 5. > "why the methods that don’t use a world model or meta-learning tend to do worse"
>
>     We have discussed in section 6 our observations and explanations for the different behaviors of the baseline methods compared to our proposed MERMAIDE algorithm. For example, in page 10, last  paragraph on “Out-of-distribution performance”, we describe that “ A world model is advantageous when 1) the test agent is more exploratory than the train set …  or 2) the agent explores throughout an episode and is likely to often select actions other than the one with its current maximum mean reward estimate …” . Would you please point us to parts that require more / better explanation so we can update accordingly?
>
>
>
>
> [1] Ruan, S., Nie, A., Steenbergen, W., He, J., Zhang, J.Q., Guo, M., Liu, Y., Nguyen, K.D., Wang, C.Y., Ying, R. and Landay, J.A., 2023. Reinforcement learning tutor better supported lower performers in a math task. arXiv preprint arXiv:2304.04933.
>
> [2] Liu, E., Stephan, M., Nie, A., Piech, C., Brunskill, E. and Finn, C., 2022. Giving Feedback on Interactive Student Programs with Meta-Exploration. Advances in Neural Information Processing Systems, 35, pp.36282-36294.
>
> [3] Hausknecht, M. and Stone, P., 2015, September. Deep recurrent q-learning for partially observable mdps. In 2015 aaai fall symposium series.
>
> [4] Chen, C., Wu, Y.F., Yoon, J. and Ahn, S., 2022. Transdreamer: Reinforcement learning with transformer world models. arXiv preprint arXiv:2202.09481.
>
> [5] Micheli, V., Alonso, E. and Fleuret, F., 2022. Transformers are sample efficient world models. arXiv preprint arXiv:2209.00588.
>
> [6] Hafner, D., Lee, K.H., Fischer, I. and Abbeel, P., 2022. Deep hierarchical planning from pixels. Advances in Neural Information Processing Systems, 35, pp.26091-26104.

---

> > ### Author Response · Authors · 2023-09-20
> > **Continued response**
> >
> > 6. > Other online learning methods for the principal-agent setting
> >
> >     Prior work in online learning for the principal-agent setting has focused on computing equilibrium strategies for the principal and the agent under different assumptions like availability of complete information about the agent [7] and fixed agent types [8]. In contrast, our goal is to learn a cost-effective intervention policy considering the setting of partial observability over the agent and few-shot adaptation to unknown agent types, which motivates our use of a deep RL approach. Using function approximation we are able to estimate the principal’s uncertainty over the agent’s actions and the meta-RL framework helps learn an intervention policy parameterization that can quickly adapt to previously unseen agents at test time.
> >
> > [7] ​​Fiscko, C., Swenson, B., Kar, S. and Sinopoli, B., 2019, June. Control of parametric games. In 2019 18th European Control Conference (ECC) (pp. 1036-1042). IEEE.
> >
> > [8] Chorppath, A.K. and Alpcan, T., 2011, December. Learning user preferences in mechanism design. In 2011 50th IEEE Conference on Decision and Control and European Control Conference (pp. 5349-5355). IEEE.

---

### Review · Reviewer_jA9T · 2023-09-14

**Summary Of Contributions:**

This paper proposes using meta-learning for Stackelberg games.    Stackelberg games are a setup where there are two agents, a principal and a a follower, and the principal tries to influence the follower to adhere to a particular policy.   The principal can affect the follower's policy by modifying its reward function at every step, however each modification of the reward incurs a certain cost, and therefore reward modification (or interventions) should be performed sparingly.  The authors propose using a meta-learning approach to learn an optimal prinicpal that can adapt to various follower behaviors.  They introduce the use of a model-based meta-learning policy that explicitly learns a model of the follower's behavior, which is then fed into the prinicpal's policy as an additional observation.

The authors demonstrate that their approach is able to find Stackleberg-optimal behaviors in simple full and noisy observation setups, and that in more complicated bandit-based setups it performs better than more naive implementations.  Pertinent ablations are performed to suggest that both the MAML and model-based mechanisms are necessary to obtain good performance.

**Audience:**

Yes

**Broader Impact Concerns:**

No broader impact concerns.

**Claims And Evidence:**

Yes

**Requested Changes:**

Although not uncommon, I'm not sure a Stackelberg game is so well known that it should not be at least briefly redefined within the paper's introduction (I had to look it up to remember what it was).  There is a brief definition in Sec. 5 but placing this earlier could be nice.  I also think a brief discussion of what the Stackelberg equilibrium corresponds to would also be pertinent.

P 4, 'Within a task, the agent's learning is affected by the prinicpal's interventions that change its reward r.  This gives rise to non-stationarity in the agent's environement....these forms of distribution shift distintguish our adaptive intervention policy learning setting from most prior work in meta-learning"  Non-stationarity is present both for the prinicpal as well as the agent.  In this section it seems that you are suggesting that the agent is affected by the principal's interventions thus creating a non-stationary learning problem for the agent, but then you go on to state that this non-stationarity distinguishes this setup from other meta-learning setups.  This is a bit confusing as the meta-learning is being (IIUC) performed on the prinicpal, and not on the agent.  And the principal /also/ has a non-stationary task, but as it is phrased it seems to suggest that it is the agent's non-stationarity which makes this meta-learning setup novel.  Can you clarify?

On page 5, after each episode of the principal's adaptation, the agent is reset, what do you mean by this?  The agent's episode is terminated or the agent is reset to an untrained state?  Please clarify here as well :)

Algorithm 1 has notations (such as mu) that seem to be only defined in Table 4, the reader should be able to fully interpret paper contents without referring to the appendix (esp. for key notations).

Sec. 5, the C/D vs NI/IN matrix is a bit confusing.  Are the 'u,1' tuples corresponding to reward and intervention cost?

I am pretty confused by the setup where the agent cannot see the principal's interventions

P. 7 "Given this equilibirium analysis....the quality of initilaization learnt using MAML than RL for adaptation.." this sentence might be missing some words? I was not able to make sense of it.

p. 11 "Our results when training on $\eps = 0.5$-greed agent show that MF-RL nad MF-MAML principals stop intervening - is this illustrated anywhere?

**Strengths And Weaknesses:**

# Strengths
This is an interesting setup, and is motivated by pertinent scenarios in governmental policy that the reviewer finds important and novel in terms of applications of RL.  The paper is very well written and for the most part easy to follow (corner cases are covered in the 'Requested Changes' section).

Experimental results are quite exhaustive within the scope of evaluated scenarios (bandits).

# Weaknesses
In the single-shot adaptation setting with full observability (p.7), each agent's $\mu$ is observed by the principal, it is therefore surprising that the REINFORCE agent does not quickly understand that $\mu>1/2 \rightarrow$ interventions and inversely.  Can you elaborate why you think this is not working?

I would have also liked to see experiments with actual RL followers and not just bandits.  Can you clarify why you did not include evaluations of this type?  I would assume the MERMAIDE agent could learn to influence these situations as well, is there some reason you did not believe it would work?

---

> ### Author Response · Authors · 2023-09-20
> **Response to review**
>
> We thank you for your time and thoughtful review. We address your primary concerns as follows:
>
> 1. We have updated the text in the main paper to add a brief description about Stackelberg games and Stackelberg equilibrium.
>
> 2.
> > And the principal /also/ has a non-stationary task, but as it is phrased it seems to suggest that it is the agent's non-stationarity which makes this meta-learning setup novel. Can you clarify?
>
>     We described the agent’s non-stationarity to convey that for a given agent (with fixed base rewards), the principal, which is the meta-learner, faces a non-stationary opponent. This is different from the common meta-learning formulations where each task is assumed to be fixed.
>
> 3.
> > On page 5, after each episode of the principal's adaptation, the agent is reset, what do you mean by this?
>
>      We mean that the agent's episode is terminated and it is reinitialized / reset to an untrained state.
>
> 4.
> >Algorithm 1 has notations (such as mu) that seem to be only defined in Table 4.
>
>     We have briefly described the notation by adding a sentence on page 6 of our updated paper.
>
> 5.
> > Are the 'u,1' tuples corresponding to reward and intervention cost?
>
>     (u,1) indicates that the agent gets a pay-off u and the principal gets a payoff of 1. c is the intervention cost incurred by the principal. We have described this in page 7 of the paper.
>
> 6.
> > I am pretty confused by the setup where the agent cannot see the principal's interventions.
>
>     In this setting, the agent does not observe the principal's action, so it has to estimate its true pay-off value $u$ from its received rewards. This setup was designed to be closer to the bandit experiment setting by introducing the learning agent and not allowing full observability of the pay-off matrix.
>
> 7.
> > ... this sentence might be missing some words?
>
>     We have updated this sentence in the paper.
>
> 8.
> > .... is this illustrated anywhere?
>
>     This sentence was in reference to our observations from the training curves for the MF-RL and MF-MAML principal. Unfortunately, we do not have access to those plots at the moment.
>
>
> 9.
> >  Can you elaborate why you think the REINFORCE agent is not working?
>
>     The poor performance of vanilla RL in sec 5 is because of the difference between the implementation of the RL and the MAML setup, which we briefly describe in page 21, para 2. At test time, the RL principal is allowed to train for K rounds (in the K-shot setup) and then evaluated, as a result for K=1 the RL policy is not able to learn the Stackelberg equilibrium intervention policy for the principal.
>
>
> 10.
> > I would have also liked to see experiments with actual RL followers and not just bandits. Can you clarify why you did not include evaluations of this type?
>
>     We believe that the MERMAIDE agent will extend to experiments with actual RL followers, however we might need to consider different architectures for the intervention policy and the world model (eg. transformers instead of GRUs) in the more complicated settings, therefore we leave that for future work. In our experiments in this paper, we showed that learning principal intervention policies even on bandit agents by itself is already challenging enough. Further, we focus on extensive evaluation around generalization to different agent types, rather than focusing on more complex RL agent problems.

---

### Decision · Action_Editor_UMpZ · 2023-11-01

**Recommendation:** Accept as is

**Comment:**

All reviewers found the paper interesting and worth publishing and the requested changes were made by the authors. The reviewers found the made changes mostly appropriate. The only remaining concern was regarding the comparison of the proposed approach to additional non-RL baselines for the Stackelberg game but this was also not considered as a reason for not accepting the paper. Hence, in line with the reviewers' recommendations, I am recommending acceptance of the paper.

**Audience:**

Yes, the considered setting has sensible real-world motivation which is relevant to parts of TMLR's audience and the evaluation of the proposed method is interesting as well.

**Claims And Evidence:**

Yes, the made claims about the proposed algorithms are supported by extensive experiments.